# Rethinking Branching on Exact Combinatorial Optimization Solver: The First Deep Symbolic Discovery Framework

**Yufei Kuang[1,2], Jie Wang[*1,2], Haoyang Liu[1,2], Fangzhou Zhu[3], Xijun Li[1,2,3],**
**Jia Zeng[3], Jianye Hao[3], Bin Li[1,2], Feng Wu[1,2]**
[1] CAS Key Laboratory of Technology in GIPAS, University of Science and Technology of China.
[2] MoE Key Laboratory of Brain-inspired Intelligent Perception and Cognition, University of Science and Technology of China.
[3] Noah's Ark Lab, Huawei Technologies.

## Abstract

Machine learning (ML) has been shown to successfully accelerate solving NP-hard combinatorial optimization (CO) problems under the branch and bound framework. However, the high training and inference cost and limited interpretability of ML approaches severely limit their wide application to modern exact CO solvers. In contrast, human-designed policies—though widely integrated in modern CO solvers due to their compactness and reliability—can not capture data-driven patterns for higher performance. To combine the advantages of the two paradigms, we propose the first symbolic discovery framework—namely, deep symbolic discovery for exact combinatorial optimization solver (Symb4CO)—to learn high-performance symbolic policies on the branching task. Specifically, we show the potential existence of small symbolic policies empirically, employ a large neural network to search in the high-dimensional discrete space, and compile the learned symbolic policies directly for fast deployment. Experiments show that the Symb4CO learned purely CPU-based policies consistently achieve *comparable* performance to previous GPU-based state-of-the-art approaches. Furthermore, the appealing features of Symb4CO include its high training (*ten training instances*) and inference (*one CPU core*) efficiency and good interpretability (*one-line expressions*), making it simple and reliable for deployment. The results show encouraging potential for the *wide* deployment of ML to modern solvers. Codes are available at `https://github.com/MIRALab-USTC/L2O-Symb4CO`.

## 1 Introduction

Combinatorial optimization (CO) problem is one of the most fundamental and challenging optimization problems in the field of mathematical optimization (MO), widely used to formulate a rich set of important real-world problems, e.g., routing (Liu et al., 2008), scheduling (Chen, 2010), and chip design (Ma et al., 2019). Generally, the solving efficiency and solution quality of CO problems are related to enormous economic value (Chen et al., 2011), while solving CO problems is usually computationally expensive due to their NP-hard nature. Thus, the acceleration on solving CO problems plays a core role in the field of MO. Modern exact CO solvers like SCIP (Achterberg, 2007) and Gurobi (Gurobi, 2022) usually employ various human-designed heuristics, whose design requires considerable manual tuning and complex working flows. Recently, researchers apply machine learning (ML) to different solver components (Paulus & Krause, 2023; Li et al., 2024; He et al., 2014; Wang et al., 2023; Geng et al., 2023; Liu et al., 2023) to accelerate the solving process, and the results show encouraging improvements on problems with chosen implicit distributions.

However, the general limitations of ML approaches on training, inference, and interpretation (Landajuela et al., 2021) makes their wide deployment to modern CO solvers a relatively slow process for a long time. First, (L1) ML approaches usually require considerable training data to achieve superior

---

*Corresponding author: jiewangx@ustc.edu.cn

performance, while we may only have limited data due to reasons like privacy issues in real-world applications. Second, (L2) ML approaches in CO can encounter the dilemma between training accuracy (complex model) and inference efficiency (deployment device), as currently most CO solver servers for industrial purpose are purely CPU-based. Lastly, (L3) the 'black-box' nature of many ML approaches prevents further understanding on the learned policies, which engenders a sense of skepticism among many researchers in MO. Note that these limitations are usually regarded intrinsic for machine learning, but they starkly misalign with the requirements in industrial-level applications. Gupta et al. (2020) employ hybrid graph neural network (GNN) and multilayer perceptron (MLP) models to tackle (L2) on purely CPU-based devices, while the neural networks still lead to low training efficiency (L1) and interpretability (L3) for deployment.

In contrast, human-designed policies are widely incorporated in modern CO solvers (Achterberg, 2007). Generally, these policies consists of hard-coded mathematical expressions and working flows, whose design are coincident to human intuitions and thus regarded to be reliable. Note that the use of mathematical operators can be regarded as strong regularizations (Petersen, 2019; Landajuela et al., 2021). Thus, these policies are usually more compact compared to the ML learned policies. However, designing and developing these policies is extremely challenging as it requires extensive expert knowledge. Moreover, these policies are designed for generic purpose, missing data-driven patterns from specific data distribution for higher performance (Bengio et al., 2021).

In light of this, a natural idea is to combine the application reliability and superior performance of these two paradigms. Then, based on the idea of genetic programming (Poli et al., 2008), we propose the first symbolic discovery framework—namely, deep symbolic optimization for exact combinatorial optimization solvers (Symb4CO)—to search for high-performance symbolic policies on the branching task. First, we conduct preliminary experiments to show the potential existence of small symbolic branching policies. Then, we employ a deep symbolic discovery framework, which employ a large sequential model to search in the high-dimensional discrete space of symbolic expressions and use the behavioral cloning accuracy as the fitness measure. Finally, we compile the learned expressions directly for fast deployment to modern exact CO solvers.

The empirical results demonstrate the following advantages of Symb4CO: (1) Superior performance on pure CPU devices. Symb4CO learned purely CPU-based policies achieve comparable performance to GPU-based state-of-the-art (SOTA) approaches and outperform all the CPU-based branching policies. (2) High training and inference efficiency. We use only *ten* training instances to train Symb4CO to SOTA performance and observe it can achieve high performance even with *one* training instance, and the learned policies only require one CPU core for stable inference. (3) High interpretability. All the learned policies are one-line compact mathematical expressions, which are easy to deploy to the distributions of CO solver packages and can help researchers further understand and optimize the human-designed branching policies.

## 2 PRELIMINARIES

### 2.1 BRANCHING VARIABLE SELECTION IN THE BRANCH-AND-BOUND ALGORITHM

Many CO problems can be formulated as the mixed integer linear programmings (MILPs):

$$\arg\min_{\boldsymbol{x}} \boldsymbol{c}^T \boldsymbol{x} \quad \text{s.t.} \quad \boldsymbol{A}\boldsymbol{x} \leq \boldsymbol{b}, \quad \boldsymbol{x} \in \mathbb{Z}^p \times \mathbb{R}^{n-p},$$

where $\boldsymbol{c}$ is the cost, $\boldsymbol{A}$ is the constraint matrix, $\boldsymbol{b}$ is the constraint right hand side vector, and $p$ is the number of integer variables. Popular exact solvers such as SCIP (Achterberg, 2007) commonly employ the branch-and-bound (B&B) algorithm to solve MILPs. B&B algorithm operates by recursively solving a series of subproblems and organizing them as nodes within a search tree. When exploring each node of the subproblem, the solver performs the *branching policy*, where it selects an integer variable $x_i$ (branching variable) with fractional value $x_i^*$ in the LP solution. The solver then partitions the feasible region to generate two new subproblems of child nodes by adding constraints $x_i \leq \lfloor x_i^* \rfloor$ and $x_i \geq \lceil x_i^* \rceil$. After that, the solver determines a new subproblem to explore next.

Empirically, the policies for branching variable selection significantly influence the size of the search tree then the overall solving efficiency. The commonly used branching policies include the strong branching and pseudocost branching policies. The strong branching policy is known to produce the smallest search trees among all the heuristics but requires a huge amount of computation, while the

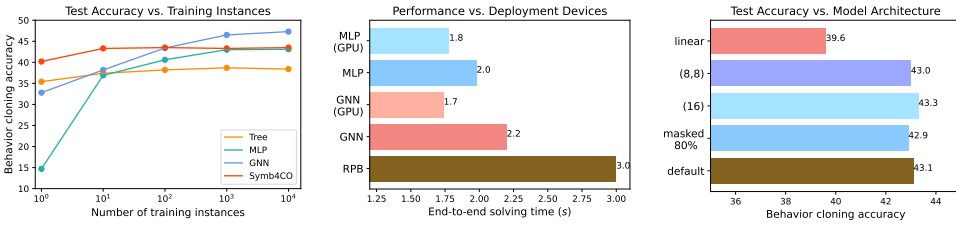

(a) Training efficiency (L1).    (b) Inference efficiency (L2).    (c) Model Architecture.

Figure 1: Motivations for symbolic discovery on the branching task. All the experiments are conducted on the combinatorial auction benchmark. Figure 1(a) shows deep learning models require extensive training data to achieve satisfactory performance, while lighter tree models struggle to achieve high learning accuracy. Figure 1(b) shows deep learning models, especially GNN, severely depend on GPU for efficient inference. Figure 1(c) shows the potential existence of compact but high-performance branching policies, which motivates us to employ symbolic operators in this work.

pseudocost branching employs a simpler and faster branching score function which depends on considerable human intuition and considerable manual tuning. Thus, these human-designed branching policies are still not satisfactory enough in real-world applications (Jünger et al., 2009), leaving vast possibilities to improve solving efficiency with ML-discovered, powerful and fast policies.

## 2.2 GENETIC PROGRAMMING FOR SYMBOLIC DISCOVERY

Genetic programming (GP) is an evolutionary computation technique that automatically solves problems without requiring the form or structure of the solution in advance. Specifically, GP approximates a mapping from the dataset by generating mathematical expressions in the form of expression trees. The internal nodes of the trees are unary or binary, representing unary (e.g. $\mathrm{pow}, \log, \exp$ and so on) and binary (e.g. $+, -, \times, \div$) mathematical operators, respectively. The leaf nodes of the trees stand for input variables and constants (e.g. variables $\{x, y, z\}$ and constants $\{c_1, c_2, c_3\}$). GP repeatedly improves the performance of the generated expressions using selection, crossover and mutation operations. Finally, a fitness metric is used for evaluating the generated expressions.

## 3 RETHINKING MACHINE LEARNING FOR BRANCHING

One of the fundamental problems in modern solvers is how to rank a set of candidates, which widely exists in modules like basis selection, cut selection, and variable selection (branching) (Jünger et al., 2009). Many policies rank candidates via human-designed scoring functions (e.g., pseudocost in branching), while designing these functions usually requires considerable expert knowledge. In this paper, we mainly focus on the branching task, in which many previous ML approaches attempt to learn better ranking functions with complex ML models like ExtraTrees (Alvarez et al., 2017), graph neural network (GNN) (Gasse et al., 2019), and hybrid models (Gupta et al., 2020).

### 3.1 LIMITATIONS FOR MACHINE LEARNING BASED BRANCHING POLICIES

Though ML based branching policies show promising results in previous research, in practice, we observe three main limitations that significantly hinder their wide deployment to modern CO solvers:

(L1) The extensive requirement on training instances. We compare the test accuracy of different ML models trained with different number of instances in Figure 1(a). Results show that limited training data severely hurts the performance of ML approaches. However, in many real-world situations we only have limited training data due to reasons like privacy issues.

(L2) The requirement for GPU devices for high performance. We compare the performance of previous SOTA approaches on CPU and GPU devices in Figure 1(b) to show that. Similar results and conclusions are reported in previous work (Gupta et al., 2020) as well. However, many servers where CO solver are deployed for industrial purpose are purely CPU-based.

(L3) The 'black-box' nature of learned models. ML models like neural networks are usually regarded uninterpretable, which engenders a sense of skepticism among MO researchers as

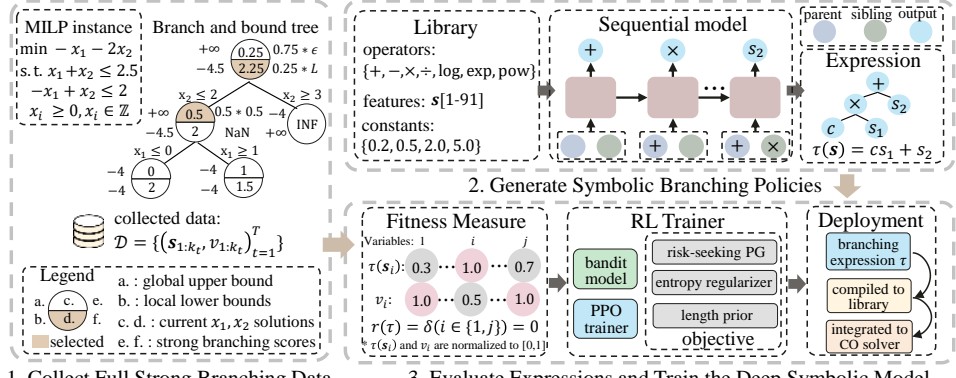

Figure 2: Illustration of the deep symbolic discovery framework for branching. Part 1 illustrate the FSB rule and the collected data with a simple MILP example; Part 2 shows how the sequential model generate symbolic policies; Part 3 shows the RL-based training process and the deployment.

> it prevents further understanding on the learned policies. Furthermore, integrating a series of complex ML models into a CO solver package distribution is typically intractable.

In conclusion, the high training and inference cost and the limited interpretability of existing ML approaches severely limit their wide deployment to modern CO solvers.

## 3.2 Motivation for Symbolic Discovery

Gupta et al. (2020) employ a hybrid approach that replaces time-consuming GNN inference with faster MLP to tackle (L2). However, the employed neural network still makes its training efficiency (L1) and interpretability (L3) unsatisfactory for practical applications. An one-step further idea is to employ lighter models to alleviate (L1) and (L3). However, previous research based on lighter SVM (Khalil et al., 2016) or tree models (Alvarez et al., 2017) falls into suboptimal learning accuracy, making it a dilemma between model size and learning capacity for these lighter models.

Thus, is there some compact model that alleviates (L1)-(L3) while maintaining a commendable level of performance? Based on the MLP model and the 91 features employed in Gupta et al. (2020), we conduct experiments and get the following results (in Figure 1(c)) to empirically answer it:

1. Sparse active features. We randomly mask $80\%$ input features and train the MLP model with remained ones. We then report the highest test accuracy among five random repeats. We find this achieves comparable accuracy to MLP trained with full features.

2. Simple but nonlinear mapping. We employ $(16)$ and $(8, 8)$ hidden layers to replace the original $(256, 256, 256)$ one used in Gupta et al. (2020). We find these compact MLP models still achieve high accuracy. However, directly removing hidden layers (i.e., using a linear model) results in clear decrease on the behavioral cloning accuracy.

We use fixed-length features rather than graph-based states in Gasse et al. (2019) as the branching policy requires to be lightweight enough for fast inference on CPU, and we use the raw features in Gupta et al. (2020) rather than the GNN embedded ones for better interpretability.

The results indicate the potential existence of compact branching policies that involve only a small set of active features and a simple (but nonlinear) mapping between inputs and decisions. Previous light ML models struggle to achieve high performance (Alvarez et al., 2017; Khalil et al., 2016; Gupta et al., 2020), which motivates us to employ more complex symbolic operators in the policies. Generally, these operators leverage the "unreasonable effectiveness of mathematics" to significantly enforce the expressive capacity of ML models (Landajuela et al., 2021).

## 4 Deep Symbolic Discovery Framework for Branching

Intuitively, the introduction of symbolic operators will make the optimization non-differentiable. That is, rather than the continuous optimization task in other ML approaches, now we need to search

for symbolic expressions in a discrete and high-dimensional space of symbolic expressions. In this section, we introduce the first symbolic discovery framework for exact combinatorial optimization solve (Symb4CO) to learn high-performance symbolic branching policies (see Figure 2).

## 4.1 THE SYMBOLIC BRANCHING FUNCTION

In genetic programming (Poli et al., 2008), functions are typically expressed as symbolic trees, in which leaf nodes are features or constants and other nodes are mathematical operators. In this task, we generate functions to score all branching candidates and select the one with the highest score.

**Input Features**. We choose the features of candidate branching variables that: a) contain as much as possible information about the branching task; b) simple enough to handle on purely CPU-based devices. Thus, we use the feature vector $s$ employed in Gupta et al. (2020), which consists of 91 features in total (see Table B8 in Appendix). These features can be roughly divided into two classes (Gasse et al., 2019; Khalil et al., 2016), i.e., the static features describing the input MILP problem and the dynamic features representing the status of the solver at current node. See Appendix E for more discussions on tackling bipartite graph Gasse et al. (2019) inputs via symbolic graph models.

**Mathematical Operators and Constants**. In the Symb4CO, we use mathematical operators from $\{+, -, \times, \div, \log, \exp, \mathrm{pow}\}$. Though there are more complex operators like $\{\sin, \cos\}$, we find their contribution to higher performance is limited in practice. We use constants from $\{0.2, 0.5, 2.0, 5.0\}$, and other constants are generated by combining them in the symbolic tree (Landajuela et al., 2021). Another approach to generate constants is to employ placeholders and optimize them in inner optimization loops at each iteration (Petersen, 2019). However, we find this approach achieves similar asymptotic performance to the above one in this task, while introducing an inner optimization loop generally requires higher training cost. See comparisons in Table 6 for comparisons.

## 4.2 THE GENERATOR OF SYMBOLIC TREES

There are multiple ways to generate the symbolic trees. Traditional approaches in genetic programming employ evolutionary algorithms (Chen et al., 2023; Cranmer et al., 2020b). In this paper, we employ deep learning models as a generator based on recent development of deep symbolic optimization algorithms (Petersen, 2019; Landajuela et al., 2021). We observe this achieves significantly higher asymptotic imitation accuracy in our task (see Table 3).

**The Sequential Model**. Once given the traversal, an expression tree is one-to-one to a expression sequence $\tau$ (Petersen, 2019). Thus, we can employ a sequential model to generate the expression sequence. Specifically, we use a recurrent neural network (RNN) to output the tree nodes step by step. At each step, we input the parent and the sibling nodes to capture the hierarchical information of the expression tree (or empty token if no parent or sibling exists) as that in Petersen (2019), and we output a categorical distribution over all possible tokens to sample the current token. Then,

$$p_{\boldsymbol{\theta}}(\tau) = \Pi_{i=1}^{|\tau|} p_{\boldsymbol{\theta}}(\tau_i \mid \tau_{1:(i-1)}) \tag{1}$$

is the likelihood to generate the expression sequence $\tau$, where $p_{\boldsymbol{\theta}}$ is approximated by an RNN.

**Constraints as Prior**. We can further apply constraints on the the search space to accelerate the training process (Petersen, 2019; Landajuela et al., 2021). In Symb4CO, we employ: a) length constraints, which restrict the complexity of the expression tree to specific ranges; b) inverse operator constraints, which restrict the child of a unary operator not to be the inverse of its parent; c) nontrivial constraints, which ensures the symbolic tree to contain at least one input feature.

## 4.3 THE FITNESS MEASURE

Due to the NP-hard nature of CO problems, training from scratch (i.e., randomly generated branching policies) with end-to-end solving time as fitness measure could be extremely intractable. Thus, similar to previous research (Khalil et al., 2016; Gasse et al., 2019; Gupta et al., 2020), we use the full strong branching (FSB) (Achterberg, 2007) to collect expert demonstrations and use the behavioral cloning accuracy as the fitness measure. The collected data $\mathcal{D} = \left\{ (\boldsymbol{s}_{1:k_t}, v_{1:k_t})_{t=1}^{T} \right\}$ includes the features $\boldsymbol{s}_i$ and the FSB scores $v_i$ of all branching candidates $i = 1, 2, \cdots, k_t$ at each node

$t = 1, 2, \cdots, T$. Then, the fitness measure can be written as

$$r(\tau) = \mathbb{E}_{(\boldsymbol{s}_{1:k}, v_{1:k}) \sim \mathcal{D}} \left[ \delta \left( \arg\max_{1:k} \tau(\boldsymbol{s}_i) \in \{i \mid v_i \geq v_j, 1 \leq j \leq k\} \right) \right], \quad (2)$$

where $(\boldsymbol{s}_{1:k}, v_{1:k})$ is an expert demonstration sampled from $\mathcal{D}$, $\delta(\mathcal{F})$ is a Dirac delta function that returns 1 if and only if the event $\mathcal{F}$ is true (Williams, 1991), and otherwise returns 0. Then, the fitness measure is that how many branching variables does the symbolic policy $\tau$ select in average over the given expert dataset $\mathcal{D}$ that have the highest FSB scores.

### 4.4 THE TRAINING ALGORITHM

Based on the generator $p_{\boldsymbol{\theta}}$ and the fitness measure $r(\tau)$, the searching task in the high-dimensional discrete space is converted to an continuous optimization on parameters $\boldsymbol{\theta}$ to maximize the probability of generating $\tau$ with high fitness. Observe that the objective is non-differentiable with respect to $\boldsymbol{\theta}$. Thus, a natural way to optimize it is to introduce reinforcement learning (RL) (Petersen, 2019).

**Problem Formulation**. Specifically, we can formulate the training task as a continuous bandit problem. The RNN generator $p_{\boldsymbol{\theta}}(\tau)$ is the parameterized policy, the expression sequence $\tau$ output by the generator is the selected action, and the fitness $r(\tau)$ with given dataset $\mathcal{D}$ is the reward.

**Policy Gradient Objective**. The general RL objective is to maximize the expected rewards of the stochastic policy $p(\tau)$, i.e., $\mathbb{E}_{\tau \sim p_{\boldsymbol{\theta}}(\tau)}[r(\tau)]$. Note that we only care about the best $\tau$ generated by the model. Thus, instead of optimizing the average performance, we employ the risk-seeking objective (Petersen, 2019; Tamar et al., 2014) to optimize the best-case performance, i.e.,

$$J(\boldsymbol{\theta}; \epsilon) = \mathbb{E}_{\tau \sim p_{\boldsymbol{\theta}}(\tau)}[r(\tau) \mid r(\tau) \geq r_\epsilon(\tau)], \quad (3)$$

where $\epsilon$ is used to select rewards larger than the $(1-\epsilon)$ quantile of all the rewards in the current batch. We employ the proximal policy optimization (PPO) algorithm (Schulman et al., 2017) to maximize this objective. We further employ the hierarchical entropy regularizer and soft length regularizer as that proposed in Landajuela et al. (2021) in Symb4CO.

### 4.5 THE DEPLOYMENT TO EXACT COMBINATORIAL OPTIMIZATION SOLVER

After the training process, we obtain a symbolic policy $\tau$ that scores all the candidate variables and then selects the best one to branch. Intuitively, this policy can be regarded as a data-driven version of the human-designed pseudocost branching rule (PB) integrated in modern CO solvers. As expected in Section 3.2, the learned policy is highly compact, which involves only a small set of active features and a simple mapping between inputs and outputs (see Table 5). Thus, similar to PB, we directly compile the learned policy to a lightweight shared object using a simple script and then integrate it into the CO solver package. See Appendix C for more details about the deployment.

## 5 RESULTS

We conduct extensive experiments to evaluate Symb4Co, which mainly have three goals: a) to illustrate that Symb4CO learned branching functions significantly outperforms existing approaches in terms of the efficiency of solving MILPs; b) to show the appealing features of Symb4CO on training, inference, and interpretability; c) to conduct ablation studies on Symb4CO.

**Benchmarks** We evaluate Symb4CO on four standard benchmarks used in previous research (Gasse et al., 2019; Gupta et al., 2020). That is, the set covering (setcover) (Balas & Ho, 1980) , the combinatorial auction (cauctions) (Leyton-Brown et al., 2000), the capacitated facility location (facilities) (Cornuéjols et al., 1991), and the maximum independent set (indset) (Bergman et al., 2015). These four benchmarks can be generated from open-source codes. We generate small instances (Easy) for training and testing, and larger instances (Medium and Hard) for generalization. We report the size of different benchmarks and the hyperparameters we used in Table D14 in Appendix.

**Baselines** There are five different baselines to compare in this section to illustrate the superior performance of GS4CO. We compare Symb4CO to seven baselines to illustrate the superior performance. Specifically, the reliability pseudocost branching (RPB), the pseudocost branching (PB), and the full

Table 1: Performance of different branching policies with the time limit of $3000s$. Both the time and the nodes are 1-shifted geometric mean over 80 test instances. All models are trained on easy instances only. (On hard benchmarks, the solver fails to obtain optimal solutions on many instances within the time limit. Thus, the node comparison may not fully reflect the actual performance.)

| Setcover: | Easy | | | Medium | | | Hard | | |
|---|---|---|---|---|---|---|---|---|---|
| Model | Time($s$) | Wins | Nodes | Time($s$) | Wins | Nodes | Time($s$) | Wins | Nodes |
| FSB | 96.96 | 0/80 | 55.2 | 242.92 | 0/80 | 107.0 | 2852.33 | 0/6 | 799.1 |
| RPB | 11.40 | 1/80 | **212.5** | 100.32 | 4/80 | 6116.0 | 1911.32 | 13/47 | 117450.0 |
| PB | 42.49 | 6/80 | 3686.6 | 112.39 | 3/80 | 9874.2 | 2158.00 | 0/20 | 149628.9 |
| Trees | 15.95 | 0/80 | 367.0 | 176.04 | 0/80 | 4350.2 | 2567.31 | 0/23 | 56962.9 |
| MLP | 9.63 | 0/80 | 290.7 | 102.63 | 0/80 | 5023.7 | 2142.81 | 0/40 | 75772.8 |
| GNN | 14.19 | 0/80 | 239.5 | 104.22 | 0/80 | **3381.5** | 2661.75 | 0/17 | **28517.8** |
| Hybrid | 8.97 | 6/80 | 279.4 | 90.20 | 8/80 | 4103.5 | 1900.96 | 6/47 | 72712.7 |
| Symb4CO | **7.10** | **67/80** | 304.7 | **86.92** | **65/80** | 5623.2 | **1894.38** | **30/48** | 129231.6 |
| GNN-GPU | 7.74 | -/80 | 239.5 | 74.91 | -/80 | 3381.5 | 1810.93 | -/50 | 92996.7 |

| Cauctions: | Easy | | | Medium | | | Hard | | |
|---|---|---|---|---|---|---|---|---|---|
| Model | Time($s$) | Wins | Nodes | Time($s$) | Wins | Nodes | Time($s$) | Wins | Nodes |
| FSB | 14.05 | 0/80 | 17.7 | 156.20 | 0/80 | 116.6 | 1903.37 | 0/51 | 533.8 |
| RPB | 3.00 | 0/80 | **26.2** | 23.05 | 1/80 | 1434.4 | 209.57 | 40/80 | **14532.4** |
| PB | 4.02 | 0/80 | 731.8 | 29.05 | 0/80 | 4855.2 | 397.38 | 2/80 | 41536.9 |
| Trees | 3.50 | 0/80 | 116.0 | 64.43 | 0/80 | 2251.6 | 1270.82 | 0/73 | 31945.8 |
| MLP | 1.98 | 0/80 | 107.8 | 21.14 | 0/80 | 1298.5 | 273.85 | 0/80 | 17295.3 |
| GNN | 2.20 | 1/80 | 93.6 | 26.93 | 0/80 | 1201.8 | 266.80 | 0/80 | 12898.0 |
| Hybrid | 1.87 | 0/80 | 96.5 | 19.04 | 4/80 | **1149.1** | 225.70 | 10/80 | 16638.3 |
| Symb4CO | **1.57** | **79/80** | 99.1 | **15.14** | **75/80** | 1328.5 | 211.08 | 28/80 | 17429.1 |
| GNN-GPU | 1.74 | -/80 | 93.6 | 16.29 | -/80 | 1298.5 | 193.14 | -/80 | 12898.0 |

| Facilities: | Easy | | | Medium | | | Hard | | |
|---|---|---|---|---|---|---|---|---|---|
| Model | Time($s$) | Wins | Nodes | Time($s$) | Wins | Nodes | Time($s$) | Wins | Nodes |
| FSB | 92.45 | 0/80 | 110.00 | 428.35 | 0/80 | 185.3 | 1089.16 | 0/69 | 76.0 |
| RPB | 54.05 | 3/80 | **211.10** | 250.88 | 1/80 | **369.3** | 766.29 | 0/78 | **270.3** |
| PB | 53.30 | 8/80 | 468.70 | 195.22 | 22/80 | 651.5 | 687.64 | 4/75 | 461.5 |
| Trees | 59.68 | 0/80 | 460.70 | 283.73 | 3/80 | 790.3 | 828.91 | 0/75 | 532.8 |
| MLP | 40.39 | 6/80 | 395.40 | 215.02 | 6/80 | 593.2 | 636.26 | 12/75 | 463.4 |
| GNN | 52.28 | 1/80 | 376.90 | 311.58 | 0/80 | 597.1 | 790.21 | 0/75 | 402.5 |
| Hybrid | 39.24 | 19/80 | 383.70 | 207.89 | 18/80 | 597.4 | 625.23 | 28/78 | 428.5 |
| Symb4CO | **37.76** | **43/80** | 401.20 | **194.01** | **30/80** | 533.6 | **618.77** | **34/78** | 377.4 |
| GNN-GPU | 34.95 | -/80 | 376.90 | 204.16 | -/80 | 597.1 | 593.39 | -/78 | 476.5 |

| Indset: | Easy | | | Medium | | | Hard | | |
|---|---|---|---|---|---|---|---|---|---|
| Model | Time($s$) | Wins | Nodes | Time($s$) | Wins | Nodes | Time($s$) | Wins | Nodes |
| FSB | 620.48 | 0/80 | 28.7 | 2110.75 | 0/41 | 133.9 | 3000.00 | 0/0 | 57.4 |
| RPB | 42.69 | 0/80 | 1500.5 | 178.43 | 10/80 | 6891.2 | 2140.77 | 5/37 | 53112.4 |
| PB | 153.59 | 1/80 | 14221.9 | 1226.90 | 0/60 | 86695.5 | 2975.40 | 0/4 | 73390.6 |
| Trees | 87.91 | 0/80 | 1317.5 | 371.75 | 0/66 | 25544.9 | 2833.05 | 0/6 | 60291.7 |
| MLP | 39.23 | 10/80 | 2490.9 | 236.63 | 10/80 | 12938.9 | 2202.30 | 2/27 | 39195.5 |
| GNN | 29.77 | 23/80 | **826.2** | 707.70 | 3/78 | 8375.2 | 2735.31 | 0/15 | 40107.2 |
| Hybrid | 36.27 | 14/80 | 2016.1 | 166.38 | 18/80 | 8337.4 | 1918.32 | 7/31 | **33558.9** |
| Symb4CO | **29.21** | **32/80** | 954.6 | **138.80** | **39/80** | **5724.2** | **1578.58** | **27/41** | 43582.6 |
| GNN-GPU | 25.13 | -/80 | 826.2 | 145.72 | -/80 | 8375.2 | 2026.70 | -/36 | 57640.6 |

strong branching (FSB) are three human-designed branching policies integrated in modern solvers like SCIP by default (Achterberg, 2007); We compare Symb4CO to seven baselines to illustrate the superior performance. Specifically, the reliability pseudocost branching (RPB), the pseudocost branching (PB), and the full strong branching (FSB) are three human-designed SOTA branching policies integrated in SCIP by default (Achterberg, 2007); the Trees model (Alvarez et al., 2017) based on the ExtraTrees (Geurts et al., 2006), the GNN (Hamilton, 2020) based model proposed in Gasse et al. (2019), and the MLP and Hybrid models proposed in Gupta et al. (2020) are four ML approaches. We compare to: a) tree models as they are widely used ML models for their reliability and interpretability; b) MLP models as Symb4CO learned policies use the same input features as that in MLP models, and thus the symbolic policies can be regarded as an extremely lightweight version of MLP; c) GNN and Hybrid models as they are SOTA ML approaches on GPU and CPU.

Table 2: The performance of ML approaches when only *one* training instance is available. Results show that Symb4CO is the only approach maintaining high performance in such condition. We note this evaluation setup is practical as we have to learn from scratch in many real-world CO tasks.

| Cauctions: | Easy | | | Medium | | | Hard | | |
|---|---|---|---|---|---|---|---|---|---|
| Model | Time($s$) | Wins | Nodes | Time($s$) | Wins | Nodes | Time($s$) | Wins | Nodes |
| RPB | 3.00 | 1/80 | **26.2** | 23.05 | 10/80 | **1434.4** | **209.57** | **65/80** | **14532.4** |
| Trees-1 | 3.13 | 0/80 | 205.6 | 67.54 | 0/80 | 2767.1 | 1692.58 | 0/51 | 56263.2 |
| MLP-1 | 3.75 | 0/80 | 541.5 | 1293.25 | 0/80 | 160516.7 | 2911.52 | 0/7 | 203175.6 |
| GNN-1 | 2.76 | 1/80 | 183.7 | 92.73 | 0/80 | 5408.4 | 1762.33 | 0/58 | 96201.0 |
| Hybrid-1 | 2.25 | 3/80 | 183.5 | 30.90 | 0/80 | 2349.2 | 691.18 | 1/49 | 38745.2 |
| Symb4CO-1 | **1.72** | **75/80** | 116.1 | **18.48** | **70/80** | 1987.7 | 255.06 | 14/80 | 20139.2 |

**Training and Evaluation Settings** For Symb4CO, we use only *ten* and four instances to generate $1,000$ and $400$ samples for training and validation on all benchmarks. We normalize the input features of all candidates inside each B&B node to $[0, 1]$ as that in previous work (Gupta et al., 2020). We observe that the end-to-end performance is not entirely positively correlated to its imitation accuracy. Thus, we select the symbolic policies with top ten imitation learning accuracy on validation sets and then execute them on the validation instances to select the best-performed one. For all the other ML-based baselines, we use their official implementations and the default settings used in previous work (Gasse et al., 2019; Gupta et al., 2020). Specifically, we use $10,000$ and $2,000$ instances to generate $100,000$ and $20,000$ samples for training and evaluation for these baselines, respectively. For Hybrid models, we use the FiLM features (Perez et al., 2018) with euclidean distance as the auxiliary task and knowledge distillation techniques, which we find effectively improves the training efficiency. We use 80 instances to evaluate the performance and the generalization ability. We report the standard metrics used in previous research: a) Time: the 1-shifted geometric mean (Achterberg, 2007) of running time in seconds; b) Nodes: the 1-shifted geometric mean of B&B nodes generated by the branching policies, which are hardware-independent; c)Wins: number of times that a branching policy wins all the others. We tune all hyperparameters on the cauctions benchmark and then directly apply to all the other benchmarks. See Appendix C for more details.

## 5.1 COMPARATIVE EVALUATION

**Solving Efficiency** We compare Symb4CO to all baselines and report the results in Table 1. Results show that Symb4CO learned purely CPU-based policies *achieve comparable performance* to GPU-based GNN policy and clearly *outperform* all the other CPU-based policies. Moreover, Symb4CO learned policies *generalized well* to Medium and Hard datasets. Note that Symb4CO achieves such performance with only *ten* training instances and *one* CPU core.

**Imitation Learning Accuracy** We report the imitation learning accuracy on the test sets in Table 3. The results show that Symb4CO learned policies achieve comparable accuracy to complex MLP models and significantly higher than the interpretable Trees. Note that Symb4CO learned policies are lightweight, interpretable, and require much fewer training instances. Symb4CO achieves higher accuracy than GPLearn (Stephens & contributors, Year of access), which is one of the SOTA open-source GP libraries.

Table 3: Imitation learning accuracy on test sets. Here GPLearn is a genetic programming library for symbolic discovery. Symb4CO achieves comparable accuracy to MLP, while it is training efficient, lightweight, and interpretable.

| Model | Cauctions | Facilities | Indset | Setcover |
|---|---|---|---|---|
| PB | 12.10 | 7.49 | 27.82 | 19.60 |
| Tree | 38.40 | 61.50 | 19.20 | 40.30 |
| MLP | 43.14 | 66.78 | 48.34 | 47.30 |
| Hybrid | 44.30 | 67.60 | 51.10 | 49.20 |
| GNN | 47.29 | 69.60 | 56.86 | 53.88 |
| GPLearn | 38.56 | 63.00 | 38.00 | 42.30 |
| Symb4CO | 43.20 | 66.30 | 46.50 | 47.70 |

## 5.2 STRENGTHS FOR APPLICATION

**Training Efficiency** Besides the results in Figure 1(a), we further compare the performance of Symb4CO to the other ML baselines with only one training instance. Results in Table 2 show that Symb4CO is significantly more training efficient than other ML approaches, as it can achieve high performance with only *one* training instance. This is mainly because that the use of symbolic

operators can be regarded as enforcing strong regularizations (Landajuela et al., 2021). We note this setting is practical as we have to learn from scratch in many real-world CO tasks.

**Inference Efficiency** We compare the decision making time (the time of feature extraction plus model inference) for different ML approaches. Results in Table 4 show that Symb4CO learned policies are extremely efficient on purely CPU-based devices. The efficiency comes from both the small set of extracted features and the compact symbolic policies. We also empirically observe that their inference efficiency is more stable than other ML policies on workstations where CPU usage exhibits significant fluctuations due to other parallel tasks.

Table 4: The average decision making time (in millisecond) of all ML-based branching policies on different problem sizes. Results show that Symb4CO learned policies are extremely efficient for inference compared to the other ML-based approaches.

| Cauctions: | Trees | MLP | GNN | Hybrid | Symb4CO | GNN-GPU |
|---|---|---|---|---|---|---|
| Easy | 14.70 | 3.00 | 5.26 | 3.73 | 0.04 | 1.74 |
| Medium | 17.20 | 6.47 | 11.57 | 7.34 | 0.07 | 2.84 |
| Hard | 28.50 | 9.56 | 10.50 | 9.91 | 0.08 | 3.19 |

**Interpretability** We report the learned policies in Table 5. We observe that expressions containing $\log$ and $\exp$ can achieve high learning accuracy but fail to achieve the best end-to-end time efficiency during validation. As expected in Section 3.2, the symbolic policies are significantly more compact than other ML models. We note that the complied symbolic policies are only $1/80$ the size of Hybrid models in average, which highly facilitates their deployment to CO solvers. Furthermore, we found policies trained on one benchmark tend to generalize well to another benchmark (Table B). We observe RPB integrated in SCIP (Gleixner et al., 2018) uses a scoring function

Table 5: Symb4CO learned branching policies are significantly more compact than other ML models (see Table B8 for feature descriptions). They are easy to be integrated into the distributions of CO solver packages due to their lightness. We believe these policies can help MO researchers to further optimize hard-coded branching policies like RPB.

| | |
|---|---|
| Setcover | $s_{89}(4s_{10} + 7s_{35} + 5s_{37} + 3s_{38} + s_{55} + 2s_{65} + s_{57} + 2s_{71})$ |
| Cauctions | $s_{35}s_{57}s_{84}s_{85}s_{89}^3 s_{90}(s_{85} + s_{89})$ |
| Facilities | $s_{89}(4s_{23} + 3s_{25} + 2s_{26} + 2s_{34} + s_{47} + 3s_{53} + 2s_{54} + 6s_{70} + 4s_{85})$ |
| Indset | $-s_{23} + 7s_{36} - s_{52} + 5s_{56} + s_{71}$ |

with human-designed features and expressions rather than vanilla pseudocosts. The intuition behind the design of this scoring function is highly similar to the learned symbolic policies in Table 5. Thus, we believe Symb4CO can help researchers further understand and optimize these hard-coded rules.

## 5.3 ABLATION STUDY

We conduct ablation study to compare different mathematical operators and constant choices (as mentioned in Section 4.1) in Table 6. First, we compare the mathematical operators we used with a larger one containing $\{\sin, \cos, \tan\}$. Then, we compare the constant placeholder approach (Petersen, 2019) with the constants used in Symb4CO. Results show that neither of them contributes to higher performance, which aligns with our expectations. Intuitively, this is because there is no periodicity in CO solvers and scoring functions in ranking tasks are usually stable against perturbations.

Table 6: Ablation study on more mathematical operators and constant choices. Results show neither of the them contributes to higher performance. Note that both introducing more operators and optimizing constant placeholders can lead to additional training costs.

| Method | Cauctions | Facilities |
|---|---|---|
| Default | 43.3 | 67.3 |
| More Math Operators | 43.3 | 67.3 |
| Constant Placeholder | 42.8 | 67.3 |

## 6 CONCLUSION AND FUTURE WORK

In this paper, we propose the first deep symbolic discovery framework Symb4CO for exact CO solver. Experiments show that Symb4CO achieves high performance and is remarkably efficient and interpretable for deployment. Exciting avenues for future work include discovering more complex working flows (e.g., the complex working flow in the RPB policy), handling more general input data structures (e.g., sequences and graphs), and deploying Symb4CO to more components in modern solvers (e.g., the simplex pricing strategies and the primal heuristics). We firmly believe the immense potential of symbolic models in a wide range of critical real-world applications like CO solvers.

## Reproducibility

We conduct all experiments on the open-source solver SCIP Optimization Suite 6.0 (Gleixner et al., 2018). All benchmarks in this paper are widely used ones in previous research (Gasse et al., 2019; Gupta et al., 2020), which can be generated directly with open-sourced codes and hyperparameters listed in Table D14. All baselines are evaluated directly using the GitHub repository (pg2455, 2023) provided by Gupta et al. (2020) and the default settings described in Section 5. The learned policies are all reported in Table 5 and the implementation details are provided in C. Then, the imitation learning accuracy can be evaluated directly using the features implemented in Gupta et al. (2020).

## Acknowledgments

The authors would like to thank all the anonymous reviewers for their insightful comments. This work was supported by National Key R&D Program of China under contract 2022ZD0119801 and National Nature Science Foundations of China grants U23A20388, U19B2026, U19B2044, and 62021001.

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

# A RELATED WORK

## A.1 MACHINE LEARNING FOR COMBINATORIAL OPTIMIZATION

Existing research on machine learning for combinatorial optimization (ML4CO) in an exact solver mainly concentrates on leveraging deep models to (1) approximate some expert heuristics that suffer from heavy computational costs (Gasse et al., 2019; Gupta et al., 2020; Zarpellon et al., 2021; Gupta et al., 2022) or (2) improve some weak heuristics with unsatisfactory performance (Chmiela et al., 2021; Wang et al., 2023). For the former category, He et al. (2014) learn a node selection policy by imitating the optimal policy given the optimal solutions; Gasse et al. (2019) and Gupta et al. (2022) use graph neural networks as fast approximations for the time-consuming strong branching heuristics with imitation learning. For the latter, Chmiela et al. (2021) construct a schedule for a collection of primal heuristics to obtain good primal solutions and learn a good schedule via reinforcement learning; Wang et al. (2023) leverage a sequential model to learn a cut selection policy by reinforcement learning. These approaches have achieved promising improvements in solving efficiency.

## A.2 LEARNING TO BRANCH IN A BRANCH-AND-BOUND SOLVER

During the solving process, branching policy is a key factor that influences the efficiency of the B&B solver. Traditional basic branching heuristics mainly include strong branching and pseudocost branching policies. The strong branching policy can produce the smallest B&B search trees but suffers from high computational costs. The pseducost branching policy uses a much simpler pseducost score to evaluate the progress of each branching candidate but requires extensive engineering experience and significant manual tuning.

Recent works have leveraged machine learning to approximate the time-consuming traditional branching policy and speed up the branching module in a B&B solver. (Khalil et al., 2016) formulate the branching variable selection problem as a ranking problem and train a variant of support vector machine (SVM) on the data collected from a strong branching expert for ranking. (Alvarez et al., 2017) uses ExtraTrees model to predict the branching scores. To capture the rich feature information in the input data, researchers use deep models with GPU inference to further enhance the solving performance. Based on the imitation-learning-based approach, (Gasse et al., 2019) represent each subproblem in the solving process as a bipartite graph and propose a graph convolutional neural network (GCNN) model to mimic the strong branching policy; (Zarpellon et al., 2021) imitates the default branching policy (reliability pseudocost branching) with deep neural network and incorporates an explicit parameterization of the state of the search tree with extra gating layers. For the reinforcement-learning-based methods, (Etheve et al., 2020) first leverage reinforcement learning for branching with deep Q-learning. Other works utilize the GCNN to parameterize the value function or policy function ((Sun et al., 2020), (Scavuzzo et al., 2022) and (Parsonson et al., 2023)). Though the existing works we mentioned above achieve high solving efficiency, they employ deep models and rely heavily on GPUs for fast inference.

However, modern exact CO solvers are usually deployed on CPU clusters. While existing CPU-friendly machine learning models such as Trees (Alvarez et al., 2017) suffer from unsatisfactory variable selection accuracy, the inference time of the deep models may become much longer on the CPU machines. Thus, the lightweight deployment of the deep models on CPUs becomes a significant topic (Gupta et al., 2020). To run the deep branching policy on the CPU machines more efficiently, Gupta et al. (2020) propose a hybrid branching policy that uses an expressive graph neural network to branch at the root node of the branch-and-bound tree and a computational-friendly multi-layer perceptron at the other nodes. To step further, Symb4CO avoids running deep models on CPU machines in the inference time and leverages the expressive deep models to search the simple symbolic mathematical expressions, in which we can achieve high inference efficiency.

## A.3 MACHINE LEARNING FOR ALGORITHM DISCOVERY

Machine learning has the potential to discover implicit rules that are beyond human intuition from training data and construct algorithms that outperform handcraft programs. ML for algorithm discovery includes symbolic discovery, program search, etc. Specifically, Program search focuses on optimizing the computing stream of the algorithm, such as Mankowitz et al. (2023) for discovering

Table B7: The cross-benchmark evaluation for learned policies in Table 5 (Time: $s$). We surprisingly found that symbolic policies trained on one benchmark can usually generalize well to another. A potential reason is that the learned symbolic policies are very compact and their parameters are very sparse, which effectively improves the generalization ability of these policies.

| Benchmark\Policy From | SetCover | Cauctions | Facilities | Indset | RPB (Default) |
|---|---|---|---|---|---|
| Setcover | **7.10** | 21.90 | **9.13** | 12.10 | 11.40 |
| Cauctions | **1.81** | **1.57** | **1.82** | **2.36** | 3.00 |
| Facilities | **46.44** | 65.07 | **37.76** | **38.40** | 54.05 |
| Indset | **22.28** | **22.16** | 66.08 | **29.21** | 42.69 |

faster sorting algorithms and Chen et al. (2023) for searching for an efficient optimization algorithm. Compared to program search, the symbolic discovery framework aims at searching the space of small mathematical expressions instead of computing streams (Petersen et al., 2021; Landajuela et al., 2021). The symbolic discovery framework is analogous to an extreme model distillation technique, extracting knowledge from a black-box neural network to hard-code expressions. Evolutionary algorithms, including genetic programming, are traditional approaches for symbolic discovery (Poli et al., 2008). Recently, deep learning has demonstrated its powerful representational capacity, offering a new approach to solving symbolic discovery problems.

## B    INPUT FEATURES

We use the 91-dimension features in Gupta et al. (2020), which are used as the input of MLP in Gupta et al. (2020). These features can be roughly divided into two classes, i.e., the static ones that describe the MILP problem and the dynamic ones that describe the solving status. We list all the features in Table B8.

## C    IMPLEMENTATION DETALS

---
**Algorithm 1** Deep Symbolic Discovery for Exact Combinatorial Optimization Solver

---
**Input:** the sequential model $p_\theta$, the library of tokens $\mathcal{L}$, and the distribution of MILP instances $\mathcal{I}$.
*// Collect expert demonstration with strong branching:*
Initial expert demonstration buffer: $\mathcal{D} \leftarrow \emptyset$.
**while** $| \mathcal{D} | < S$ **do**
    Sample MILP instance $I \sim \mathcal{I}$, solve $I$ with strong branching.
    Collect variable features and SB scores: $\mathcal{D} \leftarrow \mathcal{D} \cup \left\{ \left( \boldsymbol{s}_{1:k_t}, v_{1:k_t} \right)_{t=1}^{T} \right\}$.
    *// Here $k_t$ is the number of branching candidates at B&B node $t$.*
**end while**
*// Train the sequential model $p_\theta$:*
**for** iteration$= 1, 2, \cdots, N$ **do**
    Sample $J$ symbolic functions using tokens in $\mathcal{L}$: $\tau_{1:J} \sim p_\theta$.
    *// Calculate the fitness measure of each symbolic function for evaluation:*
    **for** $j = 1, 2, \cdots, J$ **do**
        $r(\tau_j) = \mathbb{E}_{(\boldsymbol{s}_{1:k_t}, v_{1:k_t}) \sim \mathcal{D}} \left[ \delta \left( \arg\max_{1:k_t} \tau_j(\boldsymbol{s}_i) = \arg\max_{1:k_t} v_i \right) \right]$.
    **end for**
    Train model: update $p_\theta$ via PPO by optimizing $J(\theta; \epsilon)$.
**end for**
**Output:** Symbolic function $\tau_{\text{best}}$ with highest $r(\cdot)$ on validation set.

---

**Model Architecture and Hyperparameters** Symb4CO uses LSTM to generate symbolic branching functions. The hyperparameters of Symb4CO are listed in Table C9, which roughly follows the setting in Petersen et al. (2021); Landajuela et al. (2021). All experiments are executed on Intel Xeon Platinum 2.50GHz CPUs and NVIDIA Tesla V100 GPUs. All the other baselines are executed using the implementation in previous research Gupta et al. (2020) as mentioned in Section 5.

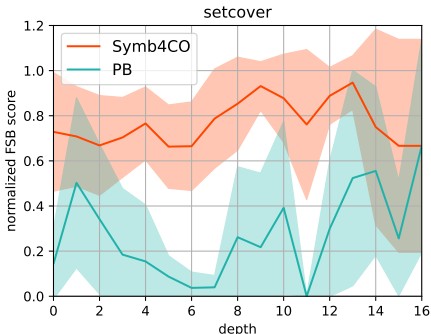

Figure C3: The curve of FSB score v.s. different depths on the Medium dataset of the Setcover benchmark.

**Constraints as Prior** There are several types of constraints employed in Symb4CO, which serves as priors to reduce the search space and accelerate the training procedure. These constraints include:

- The hard length prior, which limits the output expressions to a specific range by zeroing out the probabilities of tokens that would violate the constraint. Specifically, it sets the probabilities of leaf nodes (i.e., tokens of variables and constants whose degree of children is zero) to zero if the output expression is shorter than the minimal length and sets the probabilities of non-leaf nodes (i.e., tokens of mathematical operators) to zero if the output expression is longer than expected. We set the minimal and the maximal length to 4 and 64, respectively.

- Inverse operator constraints, which restrict the child of a unary operator not to be the inverse of its parent by zeroing out its probability. These inverse operators are the $(\exp, \log)$ pair.

- Non-trivial constraints, which ensures the symbolic tree to contain at least one input feature by zeroing out the probabilities of constants when there is only one empty leaf node left and all the other leaf nodes are constants.

- The soft length prior, which limits the lengths of the initial outputs of RNN not to concentrate at extreme points by adding a priori distribution to the outputs. Specifically, the values in the priori distribution depend on the current length of the expression, which encourages the probabilities of non-leaf nodes when the expression is shorter than the given soft length prior ($\lambda = 20$) and encourages the probabilities of leaf nodes when longer than the soft length prior. Intuitively, the motivation of the soft prior is similar to the entropy regularizer widely used in reinforcement learning.

The lengths of the learned symbolic policies are 45, 19, 55, and 49 on benchmarks Setcover, Cauctions, Facilities, and Indset, respectively. We further conduct ablation studies on the maximal hard length constraint and the soft length prior in Table C. Results show that Symb4CO is relatively insensitive to the soft length prior, while too short hard length constraint could cause decreased performance.

**Symbolic Policy Deployment** Observe that there is only a small subset of active features in the symbolic policies. Thus, we reimplement a purely C-based feature extraction code with various macros to control whether each feature should be extracted. We also switch the branching policy to RPB when depth is larger than 16 on all benchmarks to gain further acceleration, as we observe that RPB for deep nodes is both precise and efficient (see Table C, Table C, and Figure C for analysis).

## D    DATA GENERATION

We generate the dataset following the process in Gasse et al. (2019); Gupta et al. (2020), using four benchmarks of NP-hard combinatorial problem families, i.e., set covering, combinatorial auction, capacitated facility location and maximum independent set. For each benchmark, we set three levels of difficulty for instances by increasing problem scales, i.e., easy, medium and hard. For Symb4CO,

we generate *only* ten easy instances for training and four easy instances for validation. In order to obtain our datasets of state-action pairs for training and validation, we solve the training instances with SCIP and record new states and strong branching decisions during the branch-and-bound process. We collect $1,000$ samples for training, and $400$ for validation in total. For other ML-based approaches, we generate $10,000$ training instances and $2,000$ validation instances to attain $100,000$ and $20,000$ state-action pairs for training and validation on all benchmarks. All the methods are evaluated on 240 instances with the time limit of $3,000$s (80 easy instances, 80 medium instances, and 80 hard instances). The instance generation algorithms and hyperparameters for each benchmark are listed in Table D14.

## E   MORE DISCUSSIONS

**Feature Selection** Human-designed features are widely used in learning-based approaches in tasks like branching Khalil et al. (2016), node selection He et al. (2014), and cut selection Huang et al. (2022). Even the bipartite graph states employed in the branching task contain 25 human-designed features in the constraints, variables, and edges Gasse et al. (2019). In fact, during our approach design, we did consider generating features automatically from bipartite graphs based on symbolic models for graph neural networks (GNNs) Cranmer et al. (2020a); Shi et al. (2022a). However, we found three challenges that make the task non-trivial:

1. First, processing bipartite graphs via symbolic models requires complex computation by traversing the entire graph, and the complexity grows linearly with the number of layers we consider. This might be extremely expensive for inference on purely CPU-based devices compared with human-designed features.

2. Second, existing approaches learn such symbolic models by symbolize the components in GNN one by one, which results in very high training overhead compared to the lightweight Symb4CO.

3. Finally, most experiments conducted in previous research [5,6] implicitly assume the message flow passed by GNN carries a specific physical mechanism, which ensures that the message-passing function is sparse enough for symbolizing. However, the message flow in the branching task might not satisfy this assumption.

**Cross Benchmark Generalization** It is widely recognized that learning-based approaches tend to fail on out of distribution data. However, we surprisingly found that symbolic policies trained on one benchmark can usually generalize well to another benchmark (see the bold values in Table B). A potential reason is that the learned symbolic policies are very compact and their parameters are very sparse, which effectively improves the generalization ability of these policies.

**Discussions on Limitations and Future Work** We conclude three exciting challenges and their corresponding exciting future work based on the previous analysis:

1. Automated feature generation. Though the features used in this paper are simple to obtain and effective in practice, the automated feature generation is still promising future work to further help us reduce the domain knowledge for feature design, understand the underlying characteristics of this task, and deploy Symb4CO to more tasks in this field.

2. Cross-benchmark symbolic policy for general CO problems. Based on the results in Table B, we highly believe that training a cross-benchmark symbolic policy—though might not achieve the best learning accuracy on specific data distributions—to replace the SOTA human-designed scoring function in the RPB policy, is a promising avenue for future work.

3. Graph inputs and GPU-based symbolic policies. Bipartite graph is widely used to formulate a series of combinatorial optimization (CO) problems, handling these inputs is a further step towards general algorithm discovery system on CO solvers. Deploying symbolic policies on high-end GPUs (when available) can further accelerate the inference speed.

4. Deployment to satisiability (SAT) problems (Holden et al., 2021). Generally, both CO and SAT problems can be tackled via generic branch-and-bound solvers like SCIP. Thus, applying Symb4CO to SAT problems is quite an exciting and natural idea since they share many similarities in problem structures.

**Why Considering Purely CPU-Based Setting?** There are two reasons that make symbolic policies necessary:

1. Ease for real-world applications. There are two ways to distribute the learning-based CO solvers, while neither of them are GPU-friendly:

   (a) For deployment to personal users, we can not assume their accessibility to high-end GPU devices, especially for users from the field of traditional mathematical optimization.

   (b) For deployment to cloud services, we note that the inference in branching is taken at significantly high frequency. Thus, when multiple MILPs are solved in parallel, instead of packing the inputs into a batch, we have to initialize independent neural networks for each separate job to avoid the severe blocking (see results reported in Gupta et al. (2020) in Appendix 1), which results in high cost for cloud services.

2. The goal of automatic algorithm/principle discovery. Though the idea of incorporating ML to modern CO solvers is widely accepted in recent years, research that simply replaces the hard-coded components in CO solvers to "black-box" neural networks struggles to help us understand these tasks from the perspective of combinatorial optimization research. However, symbolic policies help us further understand what matters in these tasks, and thus, help researchers to further discover new algorithms/principles for these problems.

Other reasons why symbolic policies without GPU are necessary are reported in our main paper. In conclusion, compared to previous research, our work provides an entirely new perspective of the research in machine learning for combinatorial optimization (ML4CO). We believe our work will strongly contribute to the wide application of learning-based approaches to modern CO solvers.

**Tackling Bipartite Graph Inputs** Bipartite graph is a general way to represent MILP problems. Thus, a natural idea for more general purpose is to design symbolic policies that directly tackle the MILP problems with their bipartite graph representations. Specifically, following the GNN distillation approaches proposed by Cranmer et al. (2020b); Shi et al. (2022b). Then, the deployment process is just similar to that in our proposed Symb4CO.

Table B8: The 91 features used for branching candidates, following those proposed in previous research (Gupta et al., 2020; Gasse et al., 2019; Khalil et al., 2016).

| Index | Name | Short Description |
|---|---|---|
| 0-3 | Variable type | Binary, Integer, Continuous, Implied Integer |
| 4 | Normalized coefficient | Objective coefficient of the variable normalized by the norm of its coefficients in the constraints |
| 5 | Specified bounds | Whether the variable has a lower bound |
| 6 | Specified bounds | Whether the variable has a upper bound |
| 7 | Lower bound | Whether the variable reaches its lower bound in the current LP solution |
| 8 | Upper bound | Whether the variable reaches its upper bound in the current LP solution |
| 9 | Solution fractionality | Fractional part of the variable in the current LP solution |
| 10-13 | Categorical | Variable is at the lower bound, upper bound, between the bound or zero |
| 14 | Reduced cost | Amount by which objective coefficient of the variable should decrease so that the variable assumes a positive value in the LP solution |
| 15 | Age | Number of LP iterations since the last time the variable was basic normalized by total number of LP iterations |
| 16 | Solution value | Value of the variable in the current LP solution |
| 17 | Incumbent value | Value of the variable in the current best primal solution |
| 18 | Average incumbent value | Average value of the variable in all of the previously observed feasible primal solutions |
| 19-42 | Statistics for active constraint coefficients | An active constraint at a node LP is one which is binding with equality at the optimum. We consider 4 weighting schemes for an active constraint: unit weight, inverse of the sum of the coefficients of all variables in constraint, inverse of the sum of the coefficients of only candidate variables in constraint, dual cost of the constraint. Given the absolute value of the coefficients of the variable in the active constraints, we compute the sum, mean, stdev., max. and min. of those values, for each of the weighting schemes. We also compute the weighted number of active constraints that the variable is in, with the same 4 weightings |
| 43-49 | Statistics for constraint degrees | A dynamic variant of statistics for constraint degrees. Here, the constraint degrees are on the current node's LP. The ratios of the static mean, maximum and minimum to their dynamic counterparts are also features |
| 50-52 | Objective function coefficients | Value of the coefficient (raw, positive only, negative only) |
| 53-56 | Infeasibility statistics | Number and fraction of nodes for which applying SB to the variable led to one (two) infeasible children (during data collection) |
| 57 | Number of constraints | Number of constraints that the variable participates in (with a non-zero coefficient) |
| 58-59 | Min/max for ratios of constraint coefficients to RHS | Minimum and maximum ratios across negative right-hand-sides (RHS) |
| 60-67 | Min/max for one-to-all coefficient ratios | The statistics are over the ratios of a variable's coefficient, to the sum over all other variables' coefficients, for a given constraint. Four versions of these ratios are considered: positive coefficient to sum of positive coefficients |
| 68-69 | Min/max for ratios of constraint coefficients to RHS | Minimum and maximum ratios across positive right-hand-sides (RHS) |
| 70-74 | Pseudocosts | Upwards and downwards values, and their corresponding ratio, sum and product, weighted by the fractionality |
| 75-78 | Statistics for constraint degrees | The degree of a constraint is the number of variables that participate in it. A variable may participate in multiple constraints, and statistics over those constraints' degrees are used. The constraint degree is computed on the root LP (mean, stdev., min, max) |
| 79-88 | Statistics for constraint coefficients | A static version of variable's positive (negative) coefficients in the constraints it participates in (count, mean, stdev, min, max) |
| 89-90 | Slack and ceil distances | The distance to the nearest integer of the current value, the distance to the nearest integer that lager thab the current value |

Table C9: The hyperparameters for Symb4CO.

| Parameter | Value |
|---|---|
| LSTM layers | 2 |
| LSTM hidden size | 128 |
| Batch size | 500 |
| Risk factor | 0.2 |
| Expression minimal length | 4 |
| Expression maximum length | 64 |
| Soft length prior $\lambda$ | 20 |
| Soft length prior $\sigma^2$ | 8 |
| Hierarchical entropy regularizer $\gamma$ | 0.9 |
| PPO learning rate | 5e-5 |
| PPO entropy coefficient | 5e-2 |
| PPO epochs at each iteration | 8 |
| Optimizer | Adam (Kingma & Ba, 2015) |
| Number of training iterations for early stop | 300 |

Table C10: Ablation studies on the maximal hard length constraint and the soft length prior. Results show that Symb4CO is relatively insensitive to the soft length prior, while too short hard length constraint could cause decreased performance.

| Hyperparameters (on Setcover) | Imitation Learning Accuracy |
|---|---|
| Default | 47.7 |
| Maximal Length = 16 | 43.3 |
| Maximal Length = 32 | 45.1 |
| Maximal Length = 128 | 47.4 |
| Soft Length = 10 | 47.4 |
| Soft Length = 40 | 47.7 |

Table C11: The training cost of Symb4CO on each benchmark.

| Benchmark | Time (h) | Iterations |
|---|---|---|
| Setcover | 1.08 | 1087 |
| Cauctions | 1.62 | 1432 |
| Facilities | 1.87 | 1665 |
| Indset | 2.80 | 2312 |

Table C12: The decision time and the normalized FSB score of different policies, which accounts for our deployment of Symb4CO policies to first 17 layers.

| Model: | Symb4CO | | RPB | |
|---|---|---|---|---|
| Depth | Normalized FSB score | Decision Time (ms) | Normalized FSB score of the PB function | Decision Time (ms) |
| 0 | 0.728 | 56.8000 | 0.144 | 1300.5199 |
| 4 | 0.766 | 26.3000 | 0.154 | 571.3723 |
| 8 | 0.853 | 14.1000 | 0.262 | 52.0601 |
| 12 | 0.888 | 12.8486 | 0.309 | 27.8487 |
| 16 | 0.667 | 9.9025 | 0.667 | 8.5889 |

Table C13: The end-to-end performance of Symb4CO when employed in all layers (Symb4CO-AllLayers).

| Setcover: | Easy | | Medium | | Hard | |
|---|---|---|---|---|---|---|
| Model | Time | Nodes | Time | Nodes | Time | Nodes |
| RPB | 11.40 | **212.5** | 100.32 | 6116.0 | 1911.32 | 117450.0 |
| Symb4CO | 7.10 | 304.7 | **86.92** | 5623.2 | **1894.38** | 129231.6 |
| Symb4CO-AllLayers | **6.96** | 297.4 | 93.40 | **5587.6** | 1943.41 | 113149.5 |

Table D14: Details of instance generation algorithms.

| Benchmarks | Difficulty level | Generation algorithm | Hyperparameters |
|---|---|---|---|
| Set covering | Easy
Medium
Hard | Balas & Ho (1980) | 500 rows 1000 columns
1000 rows 1000 columns
2000 rows 1000 columns |
| Combinatorial auction | Easy
Medium
Hard | Leyton-Brown et al. (2000) | 100 items for 500 bids
200 items for 1000 bids
300 items for 1500 bids |
| Capacitated facility location | Easy
Medium
Hard | Cornuéjols et al. (1991) | 100 facilities with 100 customers
100 facilities with 200 customers
100 facilities with 400 customers |
| Maximum independent set | Easy
Medium
Hard | Bergman et al. (2015) | 750 nodes with affinity 4
1000 nodes with affinity 4
1500 nodes with affinity 4 |

