# OpenReview forum: "Rethinking Branching on Exact Combinatorial Optimization Solver: The First Deep Symbolic Discovery Framework"
_ICLR.cc/2024/Conference — ICLR 2024 poster_

### Official Review · Reviewer_hTDx · 2023-10-30

**Soundness:** 3 good
**Presentation:** 3 good
**Contribution:** 3 good
**Rating:** 6
**Confidence:** 3

**Summary:**

The paper introduces Symb4CO, the first symbolic discovery framework, to learn high-performance symbolic policies on the branching task in CO solvers. Symb4CO's CPU-based strategies match the performance of top GPU-based methods. With efficient training, quick inference, and clear interpretability, Symb4CO offers a promising way to integrate machine learning into modern CO solvers.

**Strengths:**

1. The problem the paper is trying to address is important. Indeed, we should have a light-weight machine learning approach to facilitate CO solvers. The directions that the author tries to improve, including requiring less training data and producing interpretable policies, are reasonable to me.
2. The authors of Symb4CO apply reinforcement learning to make training efficient with much less data needed.

**Weaknesses:**

1. Symb4CO, although data efficient during the training, requires feature extraction by human. I have concerns about if the extracted features are expressive enough, and how much efforts does it take for these feature extractions. Does it really the so-called training efficient?
2. From the experimental results, I can observe that GPU based GNN approach can make CO solvers perform faster than CPU based Symb4CO. The overall goal is to facilitate CO solvers in efficiency, if GPU based approach can facilitate it more, than why not prefer GPU based approach? Besides, today’s machine can make very fast GPU-CPU interactions. I think we should be open to bringing in machine learning models inside CO solvers (not just adding the learned interpretable policies inside CO solvers) and making use of GPU’s high computation to facilitate logical reasoning tools.

**Questions:**

Please refer to the questions in the weakness section

---

> ### Author Response · Authors · 2023-11-18
> **Response to Reviewer hTDx (Part 1/2)**
>
> We thank the reviewer for the positive and insightful comments. **We are actively improving our paper by your valuable comments**. We respond to each comment as follows and sincerely hope that our rebuttal could properly address your concerns. If so, **we would deeply appreciate it if you could further raise your score**. If not, please let us know your further concerns if any, and we will continue actively responding to your comments and improving our submission.
>
> ## Q1: How about the human effort for feature extractions?
>
> Very insightful question! The question is also asked by Reviewer z5TX. We think the effort for feature extractions is affordable in the branching task (and in many other tasks in the field of combinatorial optimization) due to the extensive existing research in combinatorial optimization (CO) solvers. Human-designed features are *widely* used in learning-based approaches in tasks like branching [1], node selection [2], and cut selection [3]. Even the bipartite graph states employed in the branching task contain 25 human-designed features in the constraints, variables, and edges [4].
>
> In fact, during our approach design, we did consider generating features automatically from bipartite graphs based on symbolic models for graph neural networks (GNNs) [5,6]. However, we found three challenges that make the task *non-trivial*:
> - First, processing bipartite graphs via symbolic models requires complex computation by traversing the entire graph, and the complexity grows linearly with the number of layers we consider. This might be extremely expensive for inference on purely CPU-based devices compared with human-designed features.
> - Second, existing approaches learn such symbolic models by symbolize the components in GNN one by one, which results in very high training overhead compared to the lightweight Symb4CO (see Table C11 in our Appendix for the training overhead of Symb4CO).
> - Finally, most experiments conducted in previous research [5,6] implicitly assume the message flow passed by GNN carries a specific physical mechanism, which ensures that the message-passing function is sparse enough for symbolizing. However, the message flow in the branching task might not satisfy this assumption.
>
> We note that our approach is the *first* step towards automated algorithm discovery on modern solvers. Based on your insight question, we fully agree the automated design of input features for different tasks (e.g., branching, cut selection, and primal heuristics)  is an *exciting avenue* for future work!
>
> ## Q2: Are the extracted features expressive enough?
>
> Thank you for the question. These features have been proved expressive in previous research [1,7]. The features are first proposed in [1] based on the domain knowledge and previous research in [8,9,10]. These features include static ones describing the input MILP problem and dynamic ones representing the status of the solver at current node, both of which contain human-designed structural information about the bipartite graph. Previous research uses these features to train support vector machines [1] and hybrid deep learning models [7] and achieves high performance on purely CPU-based devices. The bipartite graph features, though contain rich information, require graph neural networks (GNNs) to embed it via time-consuming message passing. Previous research [7] uses the embedded features at root nodes as the input of the hybrid model, but introducing these features might reduce the interpretability of the learned symbolic policies in our paper.
>
> ## Q3: Is it really so-called training efficient?
>
> Thank you for the question. As we compared in Figure 1(a) and Table 2 in our main paper, Symb4CO is significantly more efficient for training than all the other learning-based approaches. We use only *1‰* MILP instances to achieve performance comparable to previous state-of-the-art approaches. This is mainly because the symbolic policies are highly lightweight and the use of symbolic operators can be regarded as enforcing strong regularizations. We believe this feature of Symb4CO will *significantly* facilitate its application to real-world tasks.

---

> ### Author Response · Authors · 2023-11-18
> **Response to Reviewer hTDx (Part 2/2)**
>
> ## Q4: Further discussions on deploying ML models with GPU.
>
> Very thoughtful suggestion! We fully agree that researchers in this field should be open to the newest models and hardware, and there are many interesting recent papers aiming to do that [11,12,13,14]. In this paper, two reasons make symbolic policies necessary:
>
> -   Ease for real-world applications. There are two ways to distribute the learning-based CO solvers, while neither of them are GPU-friendly:
>     -   For deployment to personal users, we can not assume their accessibility to high-end GPU devices, especially for users from the field of traditional mathematical optimization.
>     -   For deployment to cloud services, we note that the inference in branching is taken at significantly high frequency. Thus, when multiple MILPs are solved in parallel, instead of packing the inputs into a batch, we have to initialize independent neural networks for each separate job to avoid the severe blocking (see Table below and results reported in [8]), which results in high cost for cloud services.
> -   The goal of automatical algorithm/principle discovery. Though the idea of incorporating ML to modern CO sovlers is widely accepted in recent years, research that simply replaces the hard-coded components in CO solvers to "black-box" neural networks struggles to help us understand these tasks from the perspective of combinatorial optimization research. However, symbolic policies help us further understand what matters in these tasks, and thus, help researchers to further discover new algorithms/principles for these problems.
>
> Other reasons why symbolic policies without GPU are necessary are reported in our main paper. In conclusion, compared to previous research, our work provides an entirely *new perspective* of the research in machine learning for combinatorial optimization (ML4CO). We believe our work will strongly contribute to the *wide* application of learning-based approaches to modern CO solvers.
>
> | Number of Parallel Tasks | Inference Time (ms) via Initializing Seperated GNNs on a Tesla V100 GPU | Inference Time (ms) via Batch on a Tesla V100 GPU |
> | ------------------------ | ------------------------------------------------------------ | ------------------------------------------------- |
> | 1| 5.4| 5.4|
> | 4| 7.3| 6.1|
> | 9| 12.6| 6.1|
> | 16| 15.3| 6.2|
> | 25| 19.0| 6.3|
>
>
> ## References
>
> [1] Khalil, Elias, et al. "Learning to branch in mixed integer programming." Proceedings of the AAAI Conference on Artificial Intelligence. Vol. 30. No. 1. 2016.
>
> [2] He, He, Hal Daume III, and Jason M. Eisner. "Learning to search in branch and bound algorithms." Advances in neural information processing systems 27 (2014).
>
> [3] Huang, Zeren, et al. "Learning to select cuts for efficient mixed-integer programming." Pattern Recognition 123 (2022): 108353.
>
> [4] Gasse, Maxime, et al. "Exact combinatorial optimization with graph convolutional neural networks." Advances in neural information processing systems 32 (2019).
>
> [5] Cranmer, Miles, et al. "Discovering symbolic models from deep learning with inductive biases." Advances in Neural Information Processing Systems 33 (2020): 17429-17442.
>
> [6] Shi, Hongzhi, et al. "Learning Symbolic Models for Graph-structured Physical Mechanism." The Eleventh International Conference on Learning Representations. 2022.
>
> [7] Gupta, Prateek, et al. "Hybrid models for learning to branch." Advances in neural information processing systems 33 (2020): 18087-18097.
>
> [8] Achterberg, Tobias, and Timo Berthold. "Hybrid branching." Integration of AI and OR Techniques in Constraint Programming for Combinatorial Optimization Problems: 6th International Conference, CPAIOR 2009 Pittsburgh, PA, USA, May 27-31, 2009 Proceedings 6. Springer Berlin Heidelberg, 2009.
>
> [9] Marcos Alvarez, Alejandro, Quentin Louveaux, and Louis Wehenkel. "A supervised machine learning approach to variable branching in branch-and-bound." (2014).
>
> [10] Patel, Jagat, and John W. Chinneck. "Active-constraint variable ordering for faster feasibility of mixed integer linear programs." Mathematical Programming 110 (2007): 445-474.
>
> [11] Wang, Zhihai, et al. "Learning Cut Selection for Mixed-Integer Linear Programming via Hierarchical Sequence Model." The Eleventh International Conference on Learning Representations. 2022.
>
> [12] Fan, Zhenan, et al. "Smart Initial Basis Selection for Linear Programs." International conference on machine learning. PMLR, 2023.
>
> [13] Paulus, Max B., and Andreas Krause. "Learning To Dive In Branch And Bound." *Thirty-seventh Conference on Neural Information Processing Systems*. 2023.
>
> [14] Li, Yang, et al. "From distribution learning in training to gradient search in testing for combinatorial optimization." *Thirty-seventh Conference on Neural Information Processing Systems*. 2023.

---

> > ### Comment · Reviewer_hTDx · 2023-11-21
> >
> > Thank you for your response. I think the future directions would be 1) improving the efficiency of GPU based inference, such as lightweight ML model, less frequent online inference or even offline inference, etc; 2) automating the feature extraction without human efforts. I suggest incorporating the content of your reply into the discussion section of your upcoming paper revision. I would like to keep my current score.

---

> > > ### Author Response · Authors · 2023-11-22
> > > **Response to Reviewer hTDx**
> > >
> > > Thank you for the feedback! All the discussions mentioned above have been added to the updated paper (in Appendix E More Discussions). We are happy that we were able to address all your questions and are grateful for your help in strengthening our work.

---

### Official Review · Reviewer_dfwS · 2023-11-01

**Soundness:** 3 good
**Presentation:** 3 good
**Contribution:** 3 good
**Rating:** 8
**Confidence:** 4

**Summary:**

The paper introduces an approach that combines the advantages of using machine learning (ML) and human-designed policies in solving combinatorial optimization (CO) problems. Symb4CO leverages deep symbolic optimization to learn interpretable symbolic expressions, transforming complex ML models into compact and efficient decision-making policies. The key strengths of Symb4CO lie in its efficiency, both in terms of training and inference, making it suitable for practical deployment on CPU-based devices. The paper shows that Symb4CO outperforms existing ML-based and human-designed policies.

Novelty:
The paper introduces the concept of using deep symbolic optimization for solving CO problems, specifically focusing on the branching task. This is a novel approach that combines symbolic representations with optimization tasks, making it distinct from traditional machine learning techniques applied to CO problems.It is designed to be highly efficient, particularly in terms of training and inference, and the learned symbolic policies are interpretable. The paper highlights the importance of these features for practical deployment in real-world applications. Symb4CO's ability to perform well on purely CPU-based devices is quite practical. Most previous ML approaches for CO problems require GPU acceleration, but Symb4CO demonstrates that it can achieve comparable performance with CPU-based resources. The paper explores the concept of using compact symbolic policies for CO tasks. This is in contrast to complex machine learning models, which are often resource-intensive. Symb4CO's approach leads to concise, one-line mathematical expressions for decision-making.

Impact:
The impact of this work is significant, especially within the field of combinatorial optimization and machine learning. Symb4CO's ability to achieve competitive performance on CPU-based devices with limited training data can make it accessible to a broader range of real-world applications. The emphasis on interpretability in Symb4CO's learned policies addresses a critical concern. Symb4CO's success on CPU-based devices reduces the need for GPU resources.

Quality of writing and experiments:
The quality of writing in the paper needs improvement. Here are some detailed comments:
- Section 4.3 talks about the fitness function, which is a crucial part of the training pipeline. So, it would be helpful to expand on the explanation of Equation 2.
- Since the authors use an FSB-based fitness function, why does the FSB model perform so poorly (Table 1) compared to their tool?
- Table 2 mentions the use of one training instance. How was this training instance chosen? How do the results vary if a different training instance was chosen? Providing a standard deviation in Table 2 would be helpful. Similarly, how were the ten instances chosen (Section 5.1)?
- Authors mention discovering more complex working flows like RPB can be an interesting future work. Since RPB is compared against in Table 1, why do the authors think RPB performs so poorly compared to Symb4CO?
- In Section 5, what is the reason for using 1-shifted geometric mean?
- The interpretability claim in Section 5.2 needs some clarity. E.g., "RPB ... uses a scoring function with human-designed features and expressions rather than vanilla pseudocosts." Why is this relevant to the interpretability claim?
- How do the authors ensure the output expression from the RNN (Figure 2) is syntactically correct?
- Providing a concrete example and comparing the results with some of the other tools might be helpful to improve clarity.
- RNNs are usually not considered suitable for long-term sequences. How did the authors tackle that problem? How long are the output expressions from the RNN model?
- Part 3 in Figure 2 is challenging to understand. The fitness measure can include concrete values to improve clarity.

Ablation studies explore the choice of mathematical operators and constants. Nevertheless, I believe it is important to address the following questions as well:
- Section 4.2 briefly mentions applying constraints during the training process. How is the length constraint implemented? Do the authors allow only a fixed length? Was there any ablation performed for the use of this constraint?
- Section 3.2 talks about 80% masking, but Section 4.1 mentions using all 91 features. How is the masking performed during training and evaluation across different benchmarks? Were any ablation studies performed on a different subset of features?
- How sensitive is the approach to different hyperparameter choices for the sequential model?
- Comment on the scalability of the approach. Do medium and hard problems have significantly larger search spaces?

Additional comments:
- Out of curiosity, did the authors explore the extent to which the learned symbolic policies generalize across different problem domains within combinatorial optimization? For example, test the policies trained on one benchmark on a different but related benchmark to assess their transferability.
- The paper discusses the efficiency and interpretability of Symb4CO. Can the authors provide insights into any trade-offs when prioritizing efficiency and simplicity over complex but potentially more accurate models?

Minor nitpick - Expand FSB in Figure 2 as it has not been used till page 3

Review summary:
The paper introduces Symb4CO, a novel approach for solving combinatorial optimization problems by combining machine learning and interpretable symbolic expressions. It leverages deep symbolic optimization to generate efficient and interpretable decision-making policies. Symb4CO is highly efficient for inference, making it suitable for CPU-based devices. The paper provides a comprehensive experimental evaluation, demonstrating its superiority over existing methods. The key strengths include its innovative use of deep symbolic optimization, efficiency, interpretability, and the potential for real-world applications. However, the paper could benefit from improved clarity, expanded related work discussion, and addressing potential limitations. The ablation studies exploring mathematical operators and constants are valuable, but additional questions and considerations should be addressed for a more comprehensive understanding.

**Strengths:**

- Symb4CO is shown to be efficient, both in terms of training and inference, making it suitable for deployment in real-world scenarios on CPU-based devices.

- The paper emphasizes the interpretability of Symb4CO's learned symbolic expressions.

- The paper presents a thorough experimental evaluation, comparing Symb4CO with various baselines and showcasing its superior performance in terms of training efficiency, inference efficiency, and overall effectiveness.

**Weaknesses:**

- The paper could benefit from improved clarity and organization. I will expand more on this in the following few sections.

- The discussion of related work is quite brief and could be expanded to provide a more comprehensive overview of existing approaches and how Symb4CO differentiates itself.

- While the paper discusses the limitations of previous ML approaches, it needs to address the potential limitations of Symb4CO. A dedicated section on drawbacks or areas for future improvement would enhance the paper's completeness.

**Questions:**

- Out of curiosity, did the authors explore the extent to which the learned symbolic policies generalize across different problem domains within combinatorial optimization? For example, test the policies trained on one benchmark on a different but related benchmark to assess their transferability.
- The paper discusses the efficiency and interpretability of Symb4CO. Can the authors provide insights into any trade-offs when prioritizing efficiency and simplicity over complex but potentially more accurate models?

---

> ### Author Response · Authors · 2023-11-18
> **Response to Reviewer dfwS (Part 1/5)**
>
> We deeply appreciate your very positive and insightful comments. **We are actively improving our paper by your valuable feedback**. We respond to each comment as follows and sincerely hope that our rebuttal could properly address your concerns. If so, **we would deeply appreciate it** **if you could further raise your score**. If not, please let us know your further concerns if any, and we will continue actively responding to your comments and improving our submission.
>
> ## Q1: Please provide detailed explanations of the fitness function and include concrete values to improve clarity in Figure 2 Part 3.
>
> Thank you for the constructive suggestion! We have revised the Part 3 of Figure 2 and the explanation of Equation (2) in our main paper to provide more details. Specifically, the fitness measure is simply defined as: if the branching variable selected by the symbolic policy has the highest full strong branching (FSB) score, then its fitness is $1$ on this sample. Otherwise, the fitness of this symbolic policy is $0$ on this sample. Then, the fitness of this symbolic policy is its average fitness over the given expert dataset $\mathcal{D}$. We found this simple fitness measure works well enough in this paper.
>
> ## Q2: Why does the full strong branching (FSB) policy perform so poorly?
>
> Thank you for the question. The main reason is that FSB need to branch and solve child nodes on all branching variables to calculate their FSB scores before selecting the best one, which usually produces small B&B trees [1] but suffers a significantly high computational cost. The FSB score in SCIP is typically defined as $v=\max (q^{-},\epsilon)\cdot\max(q^{+},\epsilon)$ [2], where $q^{-}, q^{+}$ are the dual bound changes of the two children compared to their parent and $\epsilon=10^{-6}$. Thus, it has to calculate $q^{+}, q^{-}$ for *each* branching variable at each B&B node repeatedly by solving the child nodes of each branching variable. The FSB policy is usually used as a strong but extremely slow policy to generate expert demonstrations.
>
> ## Q3: How was the single training instance chosen in Table 2 and how were the ten training instances chosen in Table 1? Please provide a standard deviation in Table 2.
>
> Thank you for the question. A similar question is also raised by Reviewer z5TX. We generate 10,000 Easy instances per benchmark using the official instance generation code provided in [1] to train the other ML baselines, and we randomly sample ten to train Symb4CO in Table 1 and randomly sample one to train Symb4CO-1 in Table 2. We report the performance of Symb4CO trained with only one instance sampled via three different random seeds and the corresponding learned policies in table below. Results show that Symb4CO *consistently* outperforms the human-designed state-of-the-art (SOTA) branching policy reliability pseudocost (RPB). Furthermore, we found the symbolic policies trained with only one instance, though still maintain high performance, have more complex forms than that trained with ten instances. A potential reason is that learning a sparse policy with only key features and operators is challenging when the training data is extremely insufficient.
>
> | Model           | Time (s)      | Nodes           | Expression                                                   |
> |-|-|-|-|
> | RPB| 3.00| 26.2||
> | Symb4CO-1-Seed0 | 1.72| 116.1| $$(2s_{89}+s_9 s_{17} s_{54} + s_9 + \exp(s_{16} s_{65} s_{89}))^{s_{85}}$$ |
> | Symb4CO-1-Seed1 | 2.05| 167.0| $$s_{84} + \exp(s_{89} (s_{86} + s_{89}s_{34} (s_{17} + s_{86} (s_{89} +   (s_{85} + s_{85} s_{89} ^ {2s_{85}} ) )) ))$$ |
> | Symb4CO-1-Seed2 | 1.67| 131.5| $$ s_{33}(s_{89} + s_{89} (s_{89} + \exp(s_{50} + s_{90} + s_{90}(s_{9}   s_{45} + s_{89}))^{s_{89}}) )^{s_{35}} $$ |
> | Average| 1.81$\pm$0.17 | 138. 2$\pm$21.3 ||
>
> ## Q4: Why does RPB perform so poorly compared to Symb4CO?
>
> Thoughtful question! Intuitively, the RPB policy is an enhanced version of the pseudocost branching (PB) policy, which warm started with strong branching when a variable is regarded as "unreliable" due to the insufficient updates of its pseudocost. However, this warm start is typically time-consuming as it needs to branch and take simplex iterations on child nodes for the unreliable branching variables. Instead, the learned symbolic policies are warm started in advance via historical data, and no strong branching is required during its execution. Thus, Symb4CO significantly outperforms RPB due to its high branching accuracy and the avoidance of the overhead for strong branching during the solving process.

---

> ### Author Response · Authors · 2023-11-18
> **Response to Reviewer dfwS (Part 2/5)**
>
> ## Q5: What is the reason for using 1-shifted geometric mean?
>
> Thank you for the question. The 1-shifted geometric mean is *widely* used to evaluate solvers due to its two distinct advantages [2]. First, compared with the arithmetic mean (i.e., $\text{Mean}(v_1, \ldots, v_k)=\frac{1}{k} \sum_{i=1}^k v_i$), it prevents the hard instances from dominating the results. Second, compared with the vanilla geometric mean (i.e., $\text{Mean}(v_1, \ldots, v_k)=(\prod_{i=1}^k \max \{v_i, 1\}t)^{\frac{1}{k}}$), it reduces the strong influence of very small values. Thus, it is widely used for the performance evaluation in classical mathematical optimization research [2], learning-based approaches [1,3], the famous LP and MILP benchmarks [4] released by Hans Mittelmann, etc.
>
> ## Q6: The interpretability claim in Section 5.2 needs some clarity.
>
> Thank you for the constructive suggestion! We have revised Section 5.2 in our main paper to provide more clarity. Specifically, the classical RPB policy employs the *pseudocost* as the scoring function to sort the branching variables and employs the strong branching to warm start "unreliable" variables [2]. However, SCIP 6.0 employs a complex *human-designed* scoring function rather than the vanilla pseudocost function, whose input features include the pseudocost, the conflict score, the cutoff score, the inference score, and etc (see Line 373 in branch_relpscost.c in the source codes of SCIP 6.0). We observe that the intuition behind the design of this scoring function is highly similar to that in our paper, while our symbolic policies are learned automatically without any human design. Thus, we believe the learned symbolic policies in this paper can further help researchers to refine the scoring function in RPB and other human-designed working flows in modern CO solvers.
>
> ## Q7: How to ensure the RNN outputs are syntactically correct?
>
> Insightful question! We employ multiple constraints to ensure the correctness of RNN outputs [5]. Detailed descriptions of these constraints can be found in Section 4.2, including the length constraints, the inverse operator constraints, and the non-trivial expression constraints. These constraints are enough to ensure the correctness of RNN outputs syntactically, but generally, they do not provide any guarantee of mathematical correctness. For example, we can not guarantee that the expression $\log(\boldsymbol{s}_0)$ is always mathematically correct, as the range of the input feature $\boldsymbol{s}_0$ can not be inferred by the RNN in advance. However, during the training process, these types of expressions always appear only in initial iterations, as they can not select enough correct branching variables on the training set $\mathcal{D}$ when they are mathematically incorrect.
>
> ## Q8: Please provide a concrete example and compare the results with other tools.
>
> Thank you for the suggestion. In our main paper, the illustrations on the challenges, the motivations, and the ablations are *consistently* based on the concrete benchmark combinatorial auction (Cautions). Specifically, the results in Figure 1 on Cauctions show the limitations of current learning-based approaches on training, inference, and interpretability; the results in Figure 1.(c) on Cauctions indicate the existence of compact branching policies; and the ablations in Table 6 on Cauctions show that Symb4CO is relatively stable to the different choices of mathematical operators and constants. We choose Cauctions for all illustrations since both the training (see Table C9) and the evaluation (see Table 1 and Table 2) on this benchmark is very fast.
>
> ## Q9: How long are the learned symbolic policies? Is RNN suitable for long-term expressions?
>
> This is a very thoughtful question! The lengths of the learned symbolic policies are 45, 19, 55, and 49 on benchmarks Setcover, Cauctions, Facilities, and Indset, respectively. These lengths are slightly longer than the simplified versions reported in Table 5, as expressions like $x+x+x+x+x$ can be simplified to $5x$.
>
> Consistent to the thoughtful question, recent research in the field of symbolic regression employs advanced sequential models like Transformer to learn complex expressions [6,7]. However, in this paper, we empirically found that a simple RNN performs well enough. A key result to support this claim is that the imitation learning accuracy of the symbolic policies learned by RNN is very close to that of MLP models (Table 3). Moreover, we note that rather than employing complex symbolic regression models, the main contribution of our submission is the first symbolic-based automated algorithm discovery framework on exact combinatorial optimization (CO) solvers. We fully agree refining the sequential model for more complex tasks is an *exciting avenue* for future work!

---

> ### Author Response · Authors · 2023-11-18
> **Response to Reviewer dfwS (Part 3/5)**
>
> ## Q10: How is the length constraint implemented? Do the authors allow only a fixed length? Was there any ablation performed for the use of this constraint?
>
> Thank you for raising this point. The hyperparameters used for the length constraints are reported in Table  C8 in Appendix. We further revised Appendix C to provide more details about the implementation.  Specifically, there are two types of length constraints implemented in our paper following the implementations in previous research [5,7].
>
> -   The *hard* length constraints limit the output expressions to a specific range by zeroing out the probabilities of tokens that would violate the constraint. Specifically, it sets the probabilities of leaf nodes (i.e., tokens of variables and constants whose degree of children is zero) to zero if the output expression is shorter than the minimal length and sets the probabilities of non-leaf nodes (i.e., tokens of mathematical operators) to zero if the output expression is longer than expected. We set the minimal and the maximal length to 4 and 64, respectively, in this paper.
> -   The *soft* length prior limits the lengths of the initial outputs of RNN not to concentrate at extreme points by adding a priori distribution to the outputs [7]. This priori distribution encourages non-leaf nodes when the expression is shorter than the soft length (i.e., $\lambda=20$) and encourages leaf nodes when the expression is longer than the soft length. Intuitively, the motivation of the soft prior is somewhat similar to the entropy regularizer widely used in reinforcement learning [9].
>
> We set these hyperparameters empirically without grid search, and they worked well enough in our experiments. Based on your insightful suggestion, we further conducted ablations on the maximal hard length constraint and the soft length prior and report the results below (and to our Appendix C). Results show that Symb4CO is relatively *insensitive* to the soft length prior, while too short hard length constraint could cause decreased performance.
>
> | Hyperparameters   (on Setcover) | Imitation Learning Accuracy |
> |-|-|
> | Default| 47.7|
> | Maximal Length = 16| 43.3|
> | Maximal Length = 32| 45.1|
> | Maximal Length = 128| 47.4|
> | Soft Length = 10| 47.4|
> | Soft Length = 40| 47.7|
>
> ## Q11: How is the masking performed during training and evaluation across different benchmarks?
>
> Sorry for the misunderstanding. The masking technique is only used in Section 3.2 as a preliminary experiment to analyse the underlying structure of the mapping between inputs and decisions, which motivates us to employ symbolic optimization to learn a simple and compact branching policy. Symb4CO proposed in Section 3 uses a *complete* set of branching features designed in [8] for all benchmarks, and it can select input features that are effective in different benchmarks automatically.
>
> ## Q12: How sensitive is the approach to different hyperparameter choices for the sequential model?
>
> Thank you for the question. The hyperparameters used in the sequential model are all listed in Table C8 in Appendix. They are set to the default ones employed in [5,6] without additional tuning, as we believe that the underlying branching policies are simple and compact. These default hyperparameters work well enough in our experiments, as it achieves comparable imitation learning accuracy to MLP models. These results also indicate that Symb4CO is insensitive to the hyperparameter choices of the sequential model.
>
> ##  Q13: Is Symb4CO scalable? Do medium and hard problems have significantly larger search spaces?
>
> Thank you for the question. Yes, Symb4CO is scalable to larger instances based on the experiments provided in Table 1 in our main paper. Specifically, we train Symb4CO only on Easy datasets and directly transfer them to Medium and Hard datasets. The end-to-end evaluation on time in Table 1 shows that the policies learned on Easy datasets keep high performance on larger datasets.
>
> The search space for problems of different sizes is the same, as the learned symbolic branching policy is just a scoring function for each branching variable. Different sizes of the problem only affect the number of branching variables at each B&B node, but do not affect the search space of the symbolic policies.

---

> ### Author Response · Authors · 2023-11-18
> **Response to Reviewer dfwS (Part 4/5)**
>
> ## Q14: How about the generalization ability of the symbolic policies across different benchmarks?
>
> Very **constructive** question! We did not conduct this experiment since the learned symbolic policies in Table 5 in our main paper seem to be totally different, and it is widely recognized that learning-based approaches tend to fail at out of distribution data.
>
> However, we surprisingly found that symbolic policies trained on one benchmark can usually generalize well to another benchmark! See the bold values in the table below (time: $s$), and we also added these new results to Appendix E. A potential reason is that the learned symbolic policies are very compact and their parameters are very sparse, which effectively improves the generalization ability of these policies.
>
> Based on these results and your constructive question, we *highly* believe that training a cross-benchmark symbolic policy---though might not achieve the best learning accuracy on specific data distributions---to replace the SOTA human-designed scoring function in the RPB policy, is a *very promising avenue* for future work!
>
> | Benchmark\Policy From | SetCover  | Cauctions | Facilities | Indset    | RPB (Default) |
> | - |-|-|-|-|-|
> | Setcover| **7.10**| 21.90| **9.13**| 12.10| 11.40|
> | Cauctions| **1.81**| **1.57**| **1.82**| **2.36**| 3.00|
> | Facilities| **46.44** | 65.07| **37.76**| **38.40** | 54.05|
> | Indset| **22.28** | **22.16** | 66.08| **29.21** | 42.69|
>
> ## Q15: How about the trade-off between simple and efficient models and complex but more accurate models?
>
> This is a very insightful question! A similar question is also raised by Reviewer z5TX. As described in Appendix C, we deploy the learned symbolic policies to the CO solver at depths 0-16 and switch back to the default RPB policy when depth is larger than 16 to accelerate performance on hard instances. The main reason for such implementation is the trade-off between efficiency and accuracy. Specifically, we found:
>
> -   The accuracy of pseudocost gradually increases to comparable to the learned ones at the deep layers of the tree. This is mainly because: a) the pseudocost in deep layers is *fully* updated after enough node expansions; b) the number of branching candidates is monotonically decreasing with the depth, making the task easier in deep layers.
> -   The deep layers of the B&B tree contain a large number of nodes, thus the inference efficiency is more crucial in these layers. We note that RPB policy is faster than symbolic policies in these layers as it reduces to almost purely pseudocost branching (PB) policy. In contrast, learned-based approaches still require extracting features for each branching variable. GNN&GPU-based approaches do not implement such technique due to their higher learning accuracy achieved by complex neural networks together with their faster inference speed achieved by high-end GPUs.
>
> To further illustrate these observations, we report on Setcover: a) the decision time and the normalized full strong branching (FSB) score of these policies in different depths in Table 1 below; b) the end-to-end performance of Symb4CO when employed for all layers (Symb4CO-AllLayers) in Table 2 below; c) the curve of the normalized FSB scores v.s. different depths in Figure C3 in our Appendix.
>
> | Model:| Symb4CO|| RPB||
> |-|-|-|-|-|
> | Depth/Max Number of B&B Nodes | Normalized FSB score | Decision Time (ms) | Normalized FSB score of the PB function | Decision Time (ms) |
> | 0/1| 0.728| 56.8000| 0.144| 1300.5199|
> | 4/16| 0.766| 26.3000| 0.154| 571.3723|
> | 8/256| 0.853| 14.1000| 0.262| 52.0601|
> | 12/4096| 0.888| 12.8486| 0.309| 27.8487|
> | 16/65536| 0.667| 9.9025| 0.667| 8.5889|
>
> | Setcover:| Easy| | Medium| | Hard| |
> |-|-|-|-|-|-|-|
> | Model| Time| Nodes| Time| Nodes| Time| Nodes|
> | RPB| 11.40| **212.5** | 100.32| 6116.0| 1911.32| 117450.0|
> | Symb4CO| 7.10| 304.7| **86.92** | 5623.2| **1894.38** | 129231.6|
> | Symb4CO-AllLayers | **6.96** | 297.4| 93.40| **5587.6** | 1943.41| **113149.5** |
>
> Thus, simply switching the symbolic policy to RPB is a very *simple and efficient* way to address this trade-off!
>
> A one-step further idea to address this trade-off is to employ GNNs, MLPs, and our symbolic policies together and deploy different models at different layers of the B&B tree. However, rather than employing complex implementations to achieve higher performance, our submission mainly focuses on the ease of deployment and the high interpretability. Thus, we simply employ the technique mentioned above in our paper for this trade-off.
>
> ## Q16: The abbreviation of FSB in Figure 2 Part 1.
>
> Thank you for pointing it out. We have revised this in Figure 2 of our main paper. We are improving our paper with your valuable suggestions!

---

> ### Author Response · Authors · 2023-11-18
> **Response to Reviewer dfwS (Part 5/5)**
>
> ## Q18: A dedicated section on drawbacks and future work.
>
> Thank you for this very constructive suggestion! We briefly discussed the limitations and the future work in Section 6 (Conclusion and Future Work) in our main paper. Based on the analysis in our submission and the insightful suggestions from all the reviewers, we further provide more discussions in Appendix E. Specifically, we conclude three exciting challenges and their corresponding exciting future work as follows:
>
> -   Automated feature generation. Though the features used in this paper are simple to obtain and effective in practice, the automated feature generation is still promising future work to further help us reduce the domain knowledge for feature design, understand the underlying characteristics of this task, and deploy Symb4CO to more tasks in the field of mathematical optimization.
> -   Cross-benchmark symbolic policy for general CO problems. Based on the results in Q14, we highly believe that training a cross-benchmark symbolic policy---though might not achieve the best learning accuracy on specific data distributions---to replace the human-designed general-purpose scoring function in the RPB policy, is a very promising avenue for future work.
> -   Graph inputs and GPU-based symbolic policies. Bipartite graph is widely used to formulate a series of combinatorial optimization (CO) problems, handling these inputs is a further step towards general algorithm discovery system for CO problems. Deploying symbolic policies on high-end GPUs (when available) can further accelerate the inference speed of these policies.
>
>
>
> ## References
>
> [1] Gasse, Maxime, et al. "Exact combinatorial optimization with graph convolutional neural networks." *Advances in neural information processing systems* 32 (2019).
>
> [2] Achterberg, Tobias. "Constraint integer programming." (2007).
>
> [3] Gupta, Prateek, et al. "Hybrid models for learning to branch." *Advances in neural information processing systems* 33 (2020): 18087-18097.
>
> [4] “ASU Benchmark Page.” Plato.asu.edu. Accessed 13 Nov. 2023. https://plato.asu.edu/bench.html.
>
> [5] Petersen, Brenden K., et al. "Deep symbolic regression: Recovering mathematical expressions from data via risk-seeking policy gradients." *International Conference on Learning Representations*. 2020.
>
> [6] O’Neill, Michael. "Riccardo Poli, William B. Langdon, Nicholas F. McPhee: A Field Guide to Genetic Programming: Lulu. com, 2008, 250 pp, ISBN 978-1-4092-0073-4." (2009): 229-230.
>
> [7] Landajuela, Mikel, et al. "Discovering symbolic policies with deep reinforcement learning." *International Conference on* *Machine Learning*. PMLR, 2021.
>
> [8] Khalil, Elias, et al. "Learning to branch in mixed integer programming." *Proceedings of the AAAI Conference on* *Artificial Intelligence*. Vol. 30. No. 1. 2016.
>
> [9] Haarnoja, Tuomas, et al. "Soft actor-critic: Off-policy maximum entropy deep reinforcement learning with a stochastic actor." *International conference on* *machine learning*. PMLR, 2018.

---

### Official Review · Reviewer_z5TX · 2023-11-06

**Soundness:** 3 good
**Presentation:** 3 good
**Contribution:** 3 good
**Rating:** 6
**Confidence:** 4

**Summary:**

This paper introduce a neural-symbolic approach to discover branching heuristics for MILP solving. Instead of directly using a neural network as the branching heuristics, they propose to use a neural network to predict a symbolic expression, which is more efficient to invoke online.

**Strengths:**

- The idea of not directly using a neural network for branching prediction, but rather using it to produce a symbolic expression is quite interesting and plausible.
- The paper shows a substantial performance gain over the other methods.

**Weaknesses:**

- The paper relies on manual feature extraction. This leaves open the question what are useful features, which is difficult to answer. Presumably the choice of the features are dependent on the particular benchmarks and can have significant impact on the performance. I get that the alternative deep representation makes the inference more expensive though.
- It is unclear how the training instances are selected and how similar to the test/validation instances the training instances are.

**Questions:**

- In the experiments, are the proposed branching heuristics invoked at each node, or only the top node?
- What is the cost of the training?

---

> ### Author Response · Authors · 2023-11-18
> **Response to Reviewer z5TX (Part 1/3)**
>
> We thank the reviewer for the positive and insightful comments. **We are actively improving our paper by your valuable comments**. We respond to each comment as follows and sincerely hope that our rebuttal could properly address your concerns. If so, **we would deeply appreciate it if you could further raise your score**. If not, please let us know your further concerns if any, and we will continue actively responding to your comments and improving our submission.
>
> ## Q1: Discussions on the feature selection.
>
> Thank you for the insightful question! The question is also raised by Reviewer hTDx. In fact, human-designed features are widely used in learning-based approaches in tasks like branching [1], node selection [2], and cut selection [3]. Even the bipartite graph states in the branching task contain 25 human-designed features in the nodes and edges [4]. Thanks to the extensive classical research in combinatorial optimization (CO) [5,6,7,8,9], designing useful features for each task are not so challenging, and previous learning-based research [1,10] have proved the effectiveness of the features employed in our paper. As expected by the reviewer, the useful features depend on particular benchmarks, but an appealing feature of Symb4CO is that it can automatically select a subset of useful features in its learned expressions (as reported in Table 5 in our main paper).
>
> During our approach design, we did consider generating features automatically from bipartite graphs based on symbolic models for graph neural networks (GNNs) [11,12]. However, we found three challenges that make the task *non-trivial*:
> - First, processing bipartite graphs via symbolic models requires complex computation by traversing the entire graph, and the complexity grows linearly with the number of layers we consider. This might be extremely expensive for inference on purely CPU-based devices compared with human-designed features.
> - Second, existing approaches learn such symbolic models by symbolize the components in GNN one by one, which results in very high training overhead compared to the lightweight Symb4CO.
> - Finally, most experiments conducted in previous research [11,12] implicitly assume the message flow passed by GNN carries a specific physical mechanism, which ensures that the message-passing function is sparse enough for symbolizing. However, the message flow in the branching task might not satisfy this assumption.
>
> Our approach is the *first* step towards automated algorithm discovery on modern solvers. Based on your insight comment, we fully believe that the automatic design of input features for different tasks (e.g., branching, cut selection, and primal heuristics) is an *exciting avenue* for future work!
>
> ## Q2: How are the training instances selected and what is the similarity between the training and test data?
>
> Thank you for the question. The instances are selected following the same setting in previous work [4, 10], and these training and test (Easy) datasets are widely recognized as independent and identically distributed. Moreover, for each of these benchmarks, we establish three distinct difficulty levels: Easy, Medium, and Hard, which are generated via different hyperparameter settings used in the instance generation code [4]. You can find detailed information about these hyperparameters in Table D9, Appendix D. During training, we generate 10,000 Easy instances per benchmark to train the ML baselines and randomly select ten to train Symb4CO. During validation, we generate an additional 2,000 Easy instances for the ML baselines and randomly choose four for Symb4CO. During evaluation, we generate 80 instances for each difficulty level (Easy, Medium, and Hard) in each benchmark, resulting in a total of 240 instances per benchmark. In summary, the training, validation, and evaluation instances of the same benchmark all belong to the same problem families and are generated using the same algorithm with different hyperparameters to control the levels of difficulty.

---

> > ### Author Response · Authors · 2023-11-18
> > **Response to Reviewer z5TX (Part 2/3)**
> >
> > ## Q3: Are the proposed branching policies invoked at each node or only the top node?
> >
> > Thank you for the question. As described in Appendix C, we deploy the learned symbolic policies to the CO solver at depths 0-16 and switch back to the default reliability pseudocost policy (RPB) when depth is larger than 16. Roughly, depths 0-16 can contain more than 131,000 nodes in total. Thus, almost all nodes in the Easy and the Medium datasets and a large part of nodes in the Hard dataset are expanded by the symbolic policy. We employ such implementation since:
> > - The accuracy of pseudocost gradually increases to comparable to the learned ones at the deep layers of the tree. This is mainly because: a) the pseudocost in deep layers is fully updated after enough node expansions; b) the number of branching candidates is monotonically decreasing with the depth, making the task easier in deep layers.
> > - The deep layers of the B&B tree contain a large number of nodes, and RPB policy is faster than symbolic policies in these layers as it reduces to almost purely pseudocost branching (PB) policy. In contrast, learned-based approaches still require extracting features for each branching variable. GNN&GPU-based approaches do not implement in this way due to their higher learning accuracy achieved by complex neural networks together with faster inference speed achieved by high-end hardware.
> > To further illustrate these reasons, we report: a) the decision time and the normalized full strong branching (FSB) score of these policies in Table 1 below; b) the end-to-end performance of Symb4CO when employed in all layers (Symb4CO-AllLayers) in Table 2 below; c) the curve of the FSB scores v.s. different depths in Figure C3 in Appendix.
> >
> > | Model: | Symb4CO |  | RPB |  |
> > | --- | --- | --- | --- | --- |
> > | Depth/Max Number of B&B Nodes | Normalized FSB score | Decision Time (ms) | Normalized FSB score of the PB function | Decision Time (ms) |
> > | 0/1 | 0.728 | 56.8000 | 0.144 | 1300.5199 |
> > | 4/16 | 0.766 | 26.3000 | 0.154 | 571.3723 |
> > | 8/256 | 0.853 | 14.1000 | 0.262 | 52.0601 |
> > | 12/4096 | 0.888 | 12.8486 | 0.309 | 27.8487 |
> > | 16/65536 | 0.667 | 9.9025 | 0.667 | 8.5889 |
> >
> > | Setcover: | Easy |  | Medium |  | Hard |  |
> > | --- | --- | --- | --- | --- | --- | --- |
> > | Model | Time | Nodes | Time | Nodes | Time | Nodes |
> > | RPB | 11.40 | **212.5** | 100.32 | 6116.0 | 1911.32 | 117450.0 |
> > | Symb4CO | 7.10 | 304.7 | **86.92** | 5623.2 | **1894.38** | 129231.6 |
> > | Symb4CO-AllLayers | **6.96** | 297.4 | 93.40 | **5587.6** | 1943.41 | **113149.5** |
> >
> > Thus, switching the symbolic policy to RPB can further accelerate the solving process on *hard* instances. Intuitively, we can regard the RPB policy as a specific PB policy warm started with FSB, while Symb4CO uses symbolic policies learned with historical data to warm start PB.
> >
> > ## Q4: What is the cost of the training?
> >
> > Thank you for the question. We add the training time of each benchmark in Table C9 in Appendix and report it in the table below. The training is conducted on a NVIDIA Tesla V100 GPU and early stoped if the accuracy does not improve for 300 iterations. Results show that Symb4CO can learn high-quality branching policies efficiently. Generally, deep learning (DL) approaches for symbolic discovery are slower than genetic programming approaches [14]. However, as reported in Table 3 in our main paper, our DL-based approach achieves higher asymptotic accuracy than GPLearn. Thus, we employ the DL-based approach in this paper at the cost of slightly higher training time.
> >
> > | Benchmark | Time (h) | Iterations |
> > | --- | --- | --- |
> > | Setcover | 1.08 | 1087 |
> > | Cauctions | 1.62 | 1432 |
> > | Facilities | 1.87 | 1665 |
> > | Indset | 2.80 | 2312 |

---

> > ### Author Response · Authors · 2023-11-18
> > **Response to Reviewer z5TX (Part 3/3)**
> >
> > ## References
> >
> > [1] Khalil, Elias, et al. "Learning to branch in mixed integer programming." Proceedings of the AAAI Conference on Artificial Intelligence. Vol. 30. No. 1. 2016.
> >
> > [2] He, He, Hal Daume III, and Jason M. Eisner. "Learning to search in branch and bound algorithms." Advances in neural information processing systems 27 (2014).
> >
> > [3] Huang, Zeren, et al. "Learning to select cuts for efficient mixed-integer programming." Pattern Recognition 123 (2022): 108353.
> >
> > [4] Gasse, Maxime, et al. "Exact combinatorial optimization with graph convolutional neural networks." Advances in neural information processing systems 32 (2019).
> >
> > [5] Achterberg, Tobias, and Timo Berthold. "Hybrid branching." Integration of AI and OR Techniques in Constraint Programming for Combinatorial Optimization Problems: 6th International Conference, CPAIOR 2009 Pittsburgh, PA, USA, May 27-31, 2009 Proceedings 6. Springer Berlin Heidelberg, 2009.
> >
> > [6] Marcos Alvarez, Alejandro, Quentin Louveaux, and Louis Wehenkel. "A supervised machine learning approach to variable branching in branch-and-bound." (2014).
> >
> > [7] Patel, Jagat, and John W. Chinneck. "Active-constraint variable ordering for faster feasibility of mixed integer linear programs." Mathematical Programming 110 (2007): 445-474.
> >
> > [8] Achterberg, Tobias. "Constraint integer programming." (2007).
> >
> > [9] Maros, István. Computational techniques of the simplex method. Vol. 61. Springer Science & Business Media, 2012.
> >
> > [10] Gupta, Prateek, et al. "Hybrid models for learning to branch." Advances in neural information processing systems 33 (2020): 18087-18097.
> >
> > [11] Cranmer, Miles, et al. "Discovering symbolic models from deep learning with inductive biases." Advances in Neural Information Processing Systems 33 (2020): 17429-17442.
> >
> > [12] Shi, Hongzhi, et al. "Learning Symbolic Models for Graph-structured Physical Mechanism." The Eleventh International Conference on Learning Representations. 2022.
> >
> > [13] Petersen, Brenden K., et al. "Deep symbolic regression: Recovering mathematical expressions from data via risk-seeking policy gradients." International Conference on Learning Representations. 2020.
> >
> > [14] Kamienny, Pierre-Alexandre, et al. "End-to-end symbolic regression with transformers." Advances in Neural Information Processing Systems 35 (2022): 10269-10281.

---

> > ### Comment · Reviewer_z5TX · 2023-11-21
> >
> > Thank you for the clarifications. I'd like to keep my score.

---

> > > ### Author Response · Authors · 2023-11-22
> > > **Response to Reviewer z5TX**
> > >
> > > Thank you for the feedback! We are happy that we were able to address all your questions and are grateful for your help in strengthening our work.

---

### Author Response · Authors · 2023-11-18
**Global Response**

Dear Area Chairs and Reviewers,

We thank all the three reviewers (z5TX, dfwS, and hTDx) for their highly insightful and valuable comments. We are delighted that all the reviews have expressed a positive inclination towards accepting our submission! Overall, the reviewers think our submission is "novel" (Reviewer dfwS) and "quite interesting and plausible" (Reviewer z5TX), the motivation is "reasonable" (Reviewer hTDx), the problem is "important" (Reviewer hTDx), the approach is "quite practical" (Reviewer dfwS) and is "distinct from traditional ML techniques" (Reviewer dfwS), the performance gain is "substantial" (Reviewer z5TX), the effect is "promising" (Reviewer hTDx) and "accessible to a broader range of applications" (Reviewer dfwS), and the impact of this work is "significant" (Reviewer dfwS).

In this paper, we provide an entirely *new perspective* of the research in the field of machine learning for combinatorial optimization (ML4CO) based on deep symbolic optimization. To further clarify this work, we conclude again the main contributions in our paper as follows:

-   (Training Efficiency) We use *only 1‰* MILP instances to achieve performance comparable to previous state-of-the-art approaches on the branching task.
-   (Deployment) Our approach runs on *purely CPU-based* devices, which we believe will significantly facilitate its wide deployment to modern CO solvers.
-   (Interpretability) Symb4CO is the *first* work to learn interpretable policies for branching, marking a key step towards automated algorithm discovery on modern solvers.
-   (Generalization Ability) The learned symbolic policies achieve remarkable *cross-benchmark* generalization ability due to their sparsity and compactness.

During the author response period, we are actively improving our paper by incorporating the reviewers' valuable suggestions. We response to all the questions raised by the reviewers in the individual responses below and revise our paper with *red* fonts, correspondingly. We summarize the main efforts in our rebuttal as follows:

-   We provide more details on the related work (Reviewer dfwS), the experimental settings (Reviewer dfwS), and the algorithm implementations (Reviewers z5TX and dfwS).
-   We add further discussions on the motivations (Reviewer hTDx), the intuitions for feature selection (Reviewers z5TX and hTDx) and algorithm design (Reviewer dfwS), and the experimental results (Reviewer dfwS).
-   We provide additional experimental results to further analyse the implementations (Reviewers z5TX and dfwS), the training overhead (Reviewer z5TX), the detailed performance (Reviewer dfwS), and the ablation studies (Reviewer dfwS).
-   We discuss several exciting avenues for future work, including automated feature generation (Reviewers z5TX and hTDx), GPU-accelerated symbolic policies (Reviewer hTDx), cross-benchmark models (Reviewer dfwS), and deployment to SAT solvers (Reviewer dfwS).

Best,

Authors

---

### Author Response · Authors · 2023-11-21
**We are looking forward to your further comments and/or questions.**

Dear Reviewers,

Thanks again for your positive and constructive comments, which are of great help to improve the quality of our work. As the rebuttal phase is approaching (due on November 22), we are eagerly looking forward to your further comments and/or questions.

We sincerely hope that our rebuttal has properly addressed your questions. If possible, we would deeply appreciate it if you could further raise your scores. If not, please let us know your further comments, and we will continue actively responding to your comments and improving our submission.

Best,

Authors

---

### Comment · Reviewer_dfwS · 2023-11-21
**Comparison with the use of ML for branching in SAT/SMT solvers**

Is it possible to provide some comparisons between your technique and the use of ML/RL for branching in SAT/SMT solvers?

There is a recent book on this topic that might be relevant:
* Machine Learning for Automated Theorem Proving: Learning to Solve SAT and QSAT by Sean Holden (https://www.nowpublishers.com/article/Details/MAL-081)

The above-cited book provides extensive review of the use of ML in SAT solvers, in particular branching. My question is focused on the use of ML for branching in solvers.

---

> ### Author Response · Authors · 2023-11-22
> **Response to Reviewer dfwS**
>
> Thank you so much for bringing this book to our attention! Generally, both CO and SAT problems can be tackled via generic branch-and-bound solvers like SCIP. Thus, applying Symb4CO to SAT problems is quite an exciting and natural idea since they share many similarities in problem structures.
>
> Based on literature research on [1,2,3]: we conclude three key advantages to deploy Symb4CO on SAT problems:
>
> -   (Training Efficiency) The limitation on available training data is even *more* severe for SAT problems. Previous research [4,5] tackles this challenge via pseudo-industrial SAT formula generation, but this is identified as one of the ten key challenges in propositional reasoning and search [6].
> -   (Deployment) SAT is a class of problems widely used to formulate real-world tasks like chip design, resource allocation in cloud computing, etc. In these tasks, high-end GPUs are not always available.
> -   (Generalization Ability) Compared to the CO problems considered in our paper, SAT problems usually suffers *more* severe generalization problems due to their highly non-smooth nature, even when the problem formulations are changed slightly.
>
> There are also two empirical results that indicate the potential of deploying Symb4CO to SAT problems:
>
> -   (Effectiveness of Features) As mentioned in Section 4.1 of [1], multiple human-designed features have been proposed for SAT problems, and previous research has proved their effectiveness on the branching task.
> -   (Branching Frequency) As mentioned in Section 3.3 of [2], the SAT solver executes a large number of branching iterations, whose magnitude tends to be significantly larger than that of CO problems. Thus, the highly efficient symbolic policies can further accelerate the branching process in SAT problems.
>
> However, before deploying to SAT solvers, we still need to validate the following points:
>
> -   (Difference in Underlying Mappings) Rather than simply improving the dual bounds, the goal of branching in SAT problems is usually more complex. Specifically, SAT problems tend to use branching policies that can fix more values, while UNSAT problems tend to find more conflicts. However, it is challenging to classify these two types of problems in advance, and sometimes stochastic branching policies could even perform better. Thus, the underlying scoring functions of SAT problems might be more complex than that of CO problems. To tackle this problem, combining complex GNNs and lightweight symbolic policies together for branching in SAT problems might be a practical idea.
> -   (Imitation Learning Labels) Designing a "strong branching" rule like that in CO solvers is challenging in SAT solvers, as there is no explicit dual bound changes to evaluate the branching quality. Currently, most of the branching policies used in SAT are purely empirical, but no ground-truth experts like strong branching are available, making the choice of imitation learning labels in SAT more challenging. To tackle this problem, perhaps combining Symb4CO with reinforcement learning is a promising approach.
>
> Due to the severe limitation of time, we struggled to deploy Symb4CO to SAT solvers. We have added the citation of book [1] to our revised paper, and we fully agree this is really an exciting avenue for future work!
>
> [1] Holden, Sean B. "Machine learning for automated theorem proving: Learning to solve SAT and QSAT." *Foundations and Trends® in Machine Learning* 14.6 (2021): 807-989.
>
> [2] Guo, Wenxuan, et al. "Machine learning methods in solving the boolean satisfiability problem." *Machine Intelligence* *Research* (2023): 1-16.
>
> [3] Achterberg, Tobias. "Constraint integer programming." (2007).
>
> [4] You, Jiaxuan, et al. "G2SAT: Learning to generate SAT formulas." *Advances in neural information processing systems* 32 (2019).
>
> [5] Li, Yang, et al. "HardSATGEN: Understanding the Difficulty of Hard SAT Formula Generation and A Strong Structure-Hardness-Aware Baseline." *arXiv* *preprint* *arXiv:2302.02104* (2023).
>
> [6] Kautz, H., D. McAllester, and B. Selman. "Ten challenges in propositional reasoning and search." *Proceedings of the Fifteenth International Joint Conference on* *Artificial Intelligence*. 1997.

---

### Meta-Review · Area_Chair_37ts · 2023-12-04

**Metareview:**

This work proposes learning heuristics for combinatorial solvers. Because these heuristics are queried many times, and the whole point is to accelerate search, they need to be fast, so the work proposes learning a symbolic program instead of training a neural network. This is a clever idea and it works well in practice. The only weakness of the work is that it appears to rely extensively upon hand-engineering features, so not exactly in the spirit of an International Conference on "Learning Representations", but overall, I agree with the reviewers that the work is practically and conceptually exciting.

**Justification For Why Not Higher Score:**

Although the work is very nice, it is also relatively niche for this conference (eg, not core ML/CV/NLP/...) , and as mentioned, relies heavily on extensive feature engineering.

**Justification For Why Not Lower Score:**

It solves an important problem in an interesting way

---

### Decision · Program_Chairs · 2024-01-16

Accept (poster)